# No Spurious Local Minima in a Two Hidden Unit ReLU Network

## Abstract

Deep learning models can be efficiently optimized via stochastic gradient descent, but there is little theoretical evidence to support this. A key question in optimization is to understand when the optimization landscape of a neural network is amenable to gradient-based optimization. We focus on a simple neural network: RELU network with one hidden layer consisting of two RELU units, and show that all local minimizers are global. This combined with recent work of Lee et al. (2017); Lee et al. (2016) show that gradient descent converges to the global minimizer.

## 1 Introduction

Deep learning has been used to achieve state-of-art performance on a wide variety of problems in machine learning, artificial intelligence, computer vision, and natural language processing. In all these applications, deep models often use hundreds of millions of parameters and are trained with stochastic gradient descent (or other gradient-based methods such as Adagrad (Duchi et al., 2011), Adam (Kingma and Ba, 2014)), a surprisingly simple method, and yet finds solutions with both low train and test error.

Despite the empirical success, the mathematical justification for gradient-based methods is not well-understood. Zhang et al. (2016a) empirically demonstrated that sufficiently over-parametrized networks can be efficiently optimized to near global optimality with stochastic gradient. For a two-layer network with leaky ReLU activation, Soudry and Carmon (2016) showed that gradient descent on a modified loss function can obtain a global minimum of the modified loss function; however, this does not imply reaching a global minimum of the original loss function. Under the same setting, Xie et al. (2016) showed that critical points with large "diversity" are nearly globally optimal. Choromanska et al. (2015) used several assumptions to simplify the loss function to a polynomial with *i.i.d.* Gaussian coefficients. They then showed that every local minima of the simplified loss has objective value comparable to the global minima. Kawaguchi (2016) used similar assumptions to show that all local minimum are global minimum in a nonlinear network. However the assumptions of Choromanska et al. (2015); Kawaguchi (2016) require *independent activations*, meaning that the activations of the hidden units are independent of the input and/or mutually independent, which is violated in practice.

Multiple works have been proposed to circumvent this assumption when dealing with the two-layer ReLU network $F(x; W) = \sum_{j=1}^{K} \sigma(w_j^T x)$, where $\sigma = \max(0, x)$ is the ReLU activation function. Under the realizable setting (*i.e.* the labels are generated from a network with "teaching" parameters $w^*$) and isotropic Gaussian input, Tian (2017) shows that when there is only a single ReLU node gradient descent converges to the global optimum. For $K = 2$, he conjectured that there are no spurious local minima, and provided a partial characterization of the critical point structure. With the same assumptions, Brutzkus and Globerson (2017) proved, for a two-layer ReLU network with a single non-overlapping convolutional filter, all local minimizers are global. Zhang et al. (2017a) show that for two-layer networks with non-standard activation functions that gradient descent converges to global minimizers.

In this paper, we focus on the case when $K = 2$ and prove that every local minimum is global. As in previous works (Brutzkus and Globerson, 2017; Tian, 2017; Hardt and Ma, 2016), we focus on the population loss. The ReLU function is positive homogeneous, so we can rewrite the function as $F(x; W) = v_1 \sigma(w_1^T x) + v_2 \sigma(w_2^T x)$ where $w_1$ and $w_2$ are unit vectors; for simplicity, we will

assume that $v_1 = v_2 = 1$. Using these assumptions and an additional orthogonality assumption, we prove that all local minima of the loss surface are global. Although the setting is a simplification of practical neural networks, this is a meaningful step towards understanding the success of gradient-based methods in deep learning and other non-convex optimization problems. For the non-orthogonal case, we provide a partial characterization of the critical point structure.

The paper is organized as follows: Section 2 discusses related works, and Section 3 introduces the notation and definitions. Section 4 shows our main result that all local minima are global and gives a proof sketch and the formal proofs are in Section 5. Section 6 provides some extensions to the non-orthogonal case. Section 7 presents the result of the experiments, and finally, Section 8 concludes the paper.

## 2 RELATED WORK

**Single Hidden Node Networks:** For a neural network with a single hidden unit and monotone activation function $\sigma$, numerous authors (Mei et al., 2016; Hazan et al., 2015; Kakade et al., 2011; Kalai and Sastry, 2009; Soltanolkotabi, 2017; Tian, 2017) have shown that gradient-based methods converge to the true parameter $w^*$. In the case of a single hidden unit, the loss function is weakly quasi-convex, meaning that the gradient points in the direction of $w^*$, which explains the success of gradient-based methods. For $K > 1$ hidden units, the loss function is no longer quasi-convex, so this analysis does not easily generalize. In fact, our analysis for $K = 2$ is considerably more involved, and requires analyzing the gradient and hessian simultaneously.

**Improper Learning:** On the improper learning side, Shalev-Shwartz et al. (2011) pioneered a kernel-based approach that can be used for learning a single halfspace or smoothed ReLU. This was generalized to fully-connected deep neural networks in Zhang et al. (2016b) using the recursive kernel method. Goel et al. (2016) designed a new smoothed ReLU function that is a better approximation to the ReLU. Instead of learning a neural network, these methods learn a function in a RKHS, hence improper learning. Zhang et al. (2017b) improved upon this by learning a neural network, instead of a kernel machine, via a boosting approach, and with much lower sample complexity. The disadvantages of improper learning are two-fold: 1) the sample complexity for these methods is exponentially larger than the Rademacher complexity of the network, and 2) the practical success of deep learning is intricately tied to using gradient-based training procedures, and the learnability of these networks using improper learning does not explain the success of gradient-based methods. On a related line of work, Janzamin et al. (2015) propose a method of moments estimator using tensor decomposition.

**Over-Parametrization** There have been several works on studying the effect of over-parametrization on the training of neural networks (Poston et al., 1991; Haeffele and Vidal, 2015). These results require the width of a hidden layer to be greater than the number of training samples, which is not the case for commonly used networks. Finally, Zhang et al. (2016a) empirically demonstrated that commonly used over-parametrized networks can be efficiently optimized to near global optimality with stochastic gradient descent.

**Non-Convex Optimization:** Since the loss function of neural networks is non-convex, the theory of training neural networks is closely related to the theory of non-convex optimization. Recently, there is considerable progress on convergence guarantees of first-order and second-order methods, including some applications in machine learning problems. Lee et al. (2016) and Lee et al. (2017) show gradient descent and other first-order methods converge only to local minima, and not saddle points. Jin et al. (2017) and Ge et al. (2015) show that variants of stochastic gradient method converge to local minimizers in polynomial time. Ge et al. (2016) and Ge et al. (2017) show there is no spurious local minima in matrix completion problem and non-convex low rank problems. For the phase retrieval problem, Sun et al. (2016) show that there is no spurious local minimum.

## 3 PRELIMINARIES

We study a simple two RELU hidden node network with output function

$$F(x; w) = \sigma(w_1^T x) + \sigma(w_2^T x).$$

For the duration of this paper, we will assume that $x$ is standard normal in $\mathbf{R}^n$ and all expectations are with respect to the standard normal. The population loss function is:

$$L(x, W) = \frac{1}{2}\mathbf{E}[(F(x, W) - F(x, W^*))^2]. \tag{1}$$

Define

$$g(v_1, v_2) = \mathbf{E}[\sigma(v_1^T x)\sigma(v_2^T x)], \tag{2}$$

so the loss can be rewritten as (ignoring additive constants, then multiplied by 4):

$$f(W) = \sum_{i,j \in \{1,2\}} \left(g(w_i, w_j) - 2g(w_i, w_j^*)\right). \tag{3}$$

From Brutzkus and Globerson (2017) we get

$$g(u, v) = \frac{1}{2\pi} \|u\| \|v\| \left(\sin \theta_{u,v} - (\pi - \theta_{u,v}) \cos \theta_{u,v}\right). \tag{4}$$

and

$$\frac{\partial g}{\partial u} = \frac{1}{2\pi} \|v\| \frac{u}{\|u\|} \sin \theta_{u,v} + \frac{1}{2\pi}(\pi - \theta_{u,v})v. \tag{5}$$

In this paper, we study the landscape of $f$ over the manifold $\mathcal{R} = \{\|w_1\| = \|w_2\| = 1\}$. The manifold gradient descent algorithm is:

$$x_{k+1} = P_{\mathcal{R}}(x_k - \alpha \nabla_{\mathcal{R}} f(x_k)),$$

where $P_{\mathcal{R}}$ is the orthogonal projector onto the manifold $\mathcal{R}$, and $\nabla_{\mathcal{R}}$ is the manifold gradient of $f$.

## 4 MAIN RESULT AND PROOF SKETCH

In order to analyze the global convergence of manifold gradient descent, we need a characterization of all critical points. We show that $f(W)$ have no spurious local minimizer on the manifold $\mathcal{R}$.

**Theorem 4.1.** *Assume $\|w_1^*\| = \|w_2^*\| = 1$ and $w_1^{*T} w_2^* = 0$, then there is no spurious local minimizer of the objective function (3) on the manifold $\mathcal{R} = \{\|w_1\| = \|w_2\| = 1\}$. Furthermore, every saddle point or local maximizer has a direction of negative curvature.*

The next theorem shows that manifold gradient descent with random initialization converges to the global minimizer

**Theorem 4.2.** *With probability one, manifold gradient descent will converge to the global minimizers.*

*Proof.* The objective function $f$ is infinitely differentiable on manifold $\mathcal{R}$. Using Corollary 6 of Lee et al. (2017), manifold gradient descent will converge to a local minimizer with probability one. Since the only local minima for function $f$ are $w_1 = w_1^*$, $w_2 = w_2^*$ and $w_1 = w_2^*$, $w_2 = w_1^*$, manifold gradient descent converges to the true solutions. □

*Proof of Theorem 4.1.* The proof of the main result is complicated, so let's start with a simpler case, in which both $w_1$ and $w_2$ are in $\mathrm{span}\{w_1^*, w_2^*\}$.

**Proposition 4.3.** *Assume $\|w_1^*\| = \|w_2^*\| = 1$, $w_1^{*T} w_2^* = 0$ and $w_1, w_2 \in \mathrm{span}\{w_1^*, w_2^*\}$, then there is no spurious local minimizer of the objective function (3) on the manifold $\mathcal{R} = \{\|w_1\| = \|w_2\| = 1\}$. Furthermore, every saddle point or local maximizer has a direction of negative curvature.*

The complete proof is given in Appendix B and C, so here we just give a proof sketch.

To prove this, we need some observations. The first important observation is that we are always on manifold $\{\|w_1\| = \|w_2\| = 1\}$, and for each vector in the plane with fixed norm, there is only one degree of freedom, which means we can express each vector with only one variable. Thus, we can express the vectors in polar coordinates, where $\theta_1$ and $\theta_2$ are the angles for $w_1$ and $w_2$.

The second observation is we only need to compute the gradient on the manifold and check whether it's zero. Define $m(w_1) = \sin\theta_1 \frac{\partial f}{\partial w_{11}} - \cos\theta_1 \frac{\partial f}{\partial w_{12}}$ and $m(w_2) = \sin\theta_2 \frac{\partial f}{\partial w_{21}} - \cos\theta_2 \frac{\partial f}{\partial w_{22}}$. Then for $w_1$ and $w_2$, the norm of the manifold gradients are $|m(w_1)|$ and $|m(w_2)|$. Thus, we only need to check whether the value of function $m$ is 0 and get rid of the absolute value sign.

Then we apply the polar coordinates onto the manifold gradients, and obtain:

$$m(w_2) = \frac{1}{\pi}(\pi - \theta_{w_1,w_2})\sin(\theta_2 - \theta_1) + \cos\theta_2 - \sin\theta_2 \tag{6}$$

$$+ \frac{1}{\pi}\left(\theta_{w_2,w_1^*}\sin\theta_2 - \theta_{w_2,w_2^*}\cos\theta_2\right). \tag{7}$$

The last observation we need for this theorem is that we must divide this problem into several cases because each angle in (300) is a piecewise linear function. If we discuss each case independently, the resulting functions are linear in the angles. The details are in Appendix B. After the calculation of all cases, we found the positions of all the critical points: WLOG assume $\theta_1 \leq \theta_2$, then there are four critical points in the 2D case: $(\theta_1, \theta_2) = (0, \frac{\pi}{2})$, $(\frac{\pi}{4}, \frac{\pi}{4})$, $(\frac{\pi}{4}, \frac{5\pi}{4})$ and $(\frac{5\pi}{4}, \frac{5\pi}{4})$.

After finding all the critical points, we compute the manifold Hessian matrix for those points and show that there is a direction of negative curvature. The details can be found in Appendix C.

The next step is to reduce it to a three dimensional problem. As stated in the two-dimensional case, the gradient is in $\mathrm{span}\{w_1, w_2, w_1^*, w_2^*\}$, which is four-dimensional. However, using the following lemma, we can reduce it to three dimensions and simplify the whole problem.

**Lemma 4.4.** *If $(w_1, w_2)$ is a critical point, then there exists a set of standard orthogonal basis $(e_1, e_2, e_3)$ such that $e_1 = w_1^*$, $e_2 = w_2^*$ and $w_1, w_2$ lies in $\mathrm{span}\{e_1, e_2, e_3\}$.*

Even if we simplify the problem into three dimensional case, it still seems to be impossible to identify all critical points explicitly. Our method to analyze the landscape of the loss surface is to find the properties of critical points and then show all saddle points and local maximizers have a direction of negative curvature.

The following two lemmas captures the main geometrical properties of the critical points in three dimensional case. More detailed properties are given is Section 5.2

**Lemma 4.5.**

$$\frac{\arccos(-w_{11})}{\arccos(-w_{21})} = \frac{\arccos(-w_{12})}{\arccos(-w_{22})} = -\frac{w_{23}}{w_{13}}. \tag{8}$$

The ratio in Lemma 4.5 captures an important property of all critical points. For simplicity, based on D.5, we define $k_0 = -k$, $\theta_1 = \pi - \theta_{w_2,w_1^*}$ and $\theta_2 = \pi - \theta_{w_2,w_2^*}$. Then

$$\pi - \theta_{w_1,w_1^*} = k_0\theta_1 \tag{9}$$

$$\pi - \theta_{w_1,w_2^*} = k_0\theta_2. \tag{10}$$

.

Then from the properties of $\theta_1$, $\theta_2$ and upper bound the value of $k_0$ we get

**Lemma 4.6.** $\theta_1 = \theta_2$.

That lemma shows that $w_1$ and $w_2$ must be on a plane whose projection onto $\mathrm{span}\{w_1^*, w_2^*\}$ is the bisector of $w_1^*$ and $w_2^*$. Combining this with the computation of Hessian, we conclude that we have found negative curvature for all possible critical points, which leads to the following proposition.

**Proposition 4.7.** *Assume $\|w_1^*\| = \|w_2^*\| = 1$, $w_1^{*T}w_2^* = 0$ and $\exists i \in [2]$, $w_i \notin \mathrm{span}\{w_1^*, w_2^*\}$, then there is no spurious local minimizer of the objective function (3) on the manifold $\{\|w_1\| = \|w_2\| = 1\}$. Furthermore, every saddle point or local maximizer has a direction of negative curvature.*

Combining both Propositions 4.3 and 4.7, we have proved Theorem 4.1, which is the main result of this paper. □

## 5 PROOFS

Here we provide some detailed proofs which are important for the understanding of the main theorem.

### 5.1 WHY WE ONLY NEED 3 DIMENSION

In general case, the following lemma shows we only need three dimension.

**Lemma 5.1.** *If $(w_1, w_2)$ is a critical point, then there exists a set of standard orthogonal basis $(e_1, e_2, e_3)$ such that $e_1 = w_1^*$, $e_2 = w_2^*$ and $w_1, w_2$ lies in $span\{e_1, e_2, e_3\}$.*

*Proof.* If $(w_1, w_2)$ is a critical point, then

$$(I - w_1 w_1^T)\frac{\partial f}{\partial w_1} = 0. \tag{11}$$

where matrix $(I - w_1 w_1^T)$ projects a vector onto the tangent space of $w_1$. Since

$$(I - w_1 w_1^T)w_1 = w_1 - w_1 = 0, \tag{12}$$

we get

$$(I - w_1 w_1^T)\frac{\partial f}{\partial w_1} \tag{13}$$

$$= \frac{1}{\pi}(I - w_1 w_1^T)\left((\pi - \theta_{w_1, w_2})w_2 - (\pi - \theta_{w_1, w_1^*})w_1^* - (\pi - \theta_{w_1, w_2^*})w_2^*\right), \tag{14}$$

which means that $(\pi - \theta_{w_1, w_2})w_2 - (\pi - \theta_{w_1, w_1^*})w_1^* - (\pi - \theta_{w_1, w_2^*})w_2^*$ lies in the direction of $w_1$. If $\theta_{w_1, w_2} = \pi$, i.e., $w_1 = -w_2$, then of course the four vectors have rank at most 3, so we can find the proper basis. If $\theta_{w_1, w_2} < \pi$, then we know that there exists a real number $r$ such that

$$(\pi - \theta_{w_1, w_2})w_2 - (\pi - \theta_{w_1, w_1^*})w_1^* - (\pi - \theta_{w_1, w_2^*})w_2^* + r \cdot w_1 = 0. \tag{15}$$

Since $\theta_{w_1, w_2} < \pi$, we know that the four vectors $w_1$, $w_2$, $w_1^*$ and $w_2^*$ are linear dependent. Thus, they have rank at most 3 and we can find the proper basis. $\square$

### 5.2 SOME PROPERTIES OF CRITICAL POINTS

Next we will focus on the properties of critical points. Assume $(w_1, w_2)$ is one of the critical points, from lemma D.1 we can find a set of standard orthogonal basis $(e_1, e_2, e_3)$ such that $e_1 = w_1^*$, $e_2 = w_2^*$ and $w_1, w_2$ lies in $span\{e_1, e_2, e_3\}$. Furthermore, assume $w_1 = w_{11}e_1 + w_{12}e_2 + w_{13}e_3$ and $w_2 = w_{21}e_1 + w_{22}e_2 + w_{23}e_3$, i.e., $w_1 = (w_{11}, w_{12}, w_{13})$ and $w_2 = (w_{21}, w_{22}, w_{23})$. Since we have already found out all the critical points when $w_{13} = w_{23} = 0$, in the following we assume $w_{13}^2 + w_{23}^2 \neq 0$.

First, we give the fundamental equation in our analysis.

**Lemma 5.2.**

$$\frac{\arccos(-w_{11})}{\arccos(-w_{21})} = \frac{\arccos(-w_{12})}{\arccos(-w_{22})} = -\frac{w_{23}}{w_{13}}. \tag{16}$$

*Proof.* Adapting from the proof of lemma D.4 and we know that

$$\frac{w_{21} - \frac{\pi - \theta_{w_1, w_1^*}}{\pi - \theta_{w_1, w_2}}}{w_{11}} = \frac{w_{22} - \frac{\pi - \theta_{w_1, w_2^*}}{\pi - \theta_{w_1, w_2}}}{w_{12}} = \frac{w_{23}}{w_{13}} = k. \tag{17}$$

Similarly, we have

$$\frac{w_{11} - \frac{\pi - \theta_{w_2, w_1^*}}{\pi - \theta_{w_1, w_2}}}{w_{21}} = \frac{w_{12} - \frac{\pi - \theta_{w_2, w_2^*}}{\pi - \theta_{w_1, w_2}}}{w_{22}} = \frac{w_{13}}{w_{23}} = \frac{1}{k}. \tag{18}$$

Taking the first component of (217) and (218) gives us

$$w_{21} = k \cdot w_{11} + \frac{\pi - \theta_{w_1, w_1^*}}{\pi - \theta_{w_1, w_2}} \tag{19}$$

$$w_{21} = k \cdot w_{11} - k \frac{\pi - \theta_{w_2, w_1^*}}{\pi - \theta_{w_1, w_2}}. \tag{20}$$

Thus,

$$\frac{\pi - \theta_{w_1, w_1^*}}{\pi - \theta_{w_2, w_1^*}} = -k. \tag{21}$$

Similarly, we get

$$\frac{\pi - \theta_{w_1, w_2^*}}{\pi - \theta_{w_2, w_2^*}} = -k. \tag{22}$$

Since $\forall i, j \in [2], \pi - \theta_{w_i, w_j^*} = \arccos(-\theta_{w_{ij}})$, we know that

$$\frac{\arccos(-w_{11})}{\arccos(-w_{21})} = \frac{\arccos(-w_{12})}{\arccos(-w_{22})} = -\frac{w_{23}}{w_{13}}. \tag{23}$$

$\square$

Using this equation, we obtain several properties of critical points. The following two lemmas show the basic properties of critical points in three dimensional case. Completed proofs are given in Appendix B and C.

**Lemma 5.3.** $\theta_{w_1, w_2} < \pi$.

**Lemma 5.4.** $w_{13} * w_{23} < 0$.

These two lemmas restrict the position of critical points in some specific domains.

Then we construct a new function $F$ in order to get more precise analysis. Define $k_0 = -k$, $\theta_1 = \pi - \theta_{w_2, w_1^*}$ and $\theta_2 = \pi - \theta_{w_2, w_2^*}$.

$$F(\theta) = \frac{-k_0 \theta}{k_0 \cos(k_0 \theta) + \cos(\theta)}, \tag{24}$$

From the properties of that particular function and upper bound the value of $k_0$ we get

**Lemma 5.5.** $\theta_1 = \theta_2$.

That lemma shows that $w_1$ and $w_2$ must be on a plane whose projection onto $\text{span}\{w_1^*, w_2^*\}$ is the bisector of $w_1^*$ and $w_2^*$. Although we cannot identify the critical points explicitly, we will show these geometric properties already capture the direction of negative curvature.

# 6 ANALYSIS OF CRITICAL POINTS FOR NON-ORTHOGONAL $W^*$

In this section, we partially characterize the structure of the critical points when $w_1^*, w_2^*$ are non-orthogonal, but form an acute angle. In other words, the angle between $w_1^*$ and $w_2^*$ is $\alpha \in (0, \frac{\pi}{2})$. Let us first consider the 2D cases, i.e., both $w_1$ and $w_2$ are in the span of $w_1^*$ and $w_2^*$. Similar to the original problem, after the technique of changing variables(i.e., using polar coordinates and assume $\theta_1$ and $\theta_2$ are the angles of $w_1$ and $w_2$ in polar coordinates), we divide the whole plane into 4 parts, which are the angle in $[0, \alpha], [\alpha, \pi], [\pi, \pi + \alpha]$ and $[\pi + \alpha, 2\pi)$. We have the following lemma:

**Lemma 6.1.** *Assume* $\|w_1^*\| = \|w_2^*\| = 1$, $w_1^{*T} w_2^* > 0$ *and* $w_1, w_2 \in \text{span}\{w_1^*, w_2^*\}$. *When* $w_1$ *and* $w_2$ *are in the same part(one of four parts), the only critical points except the global minima are those when both* $w_1$ *and* $w_2$ *are on the bisector of* $w_1^*$ *and* $w_2^*$.

*Proof.* The complete proof is given in appendix E, the techniques are nearly the same as things in the original problem and a bit harder, so to be brief, we omit the proof details here. $\square$

For the three-dimensional cases cases of this new problem, it's interesting that the first few lemmatas are still true. Specifically, Lemma D.1(restated as Lemma 4.4) to Lemma D.5(restated as Lemma 4.5) are still correct. The proof is very similar to the proofs of those lemmas, except we need modification to the coefficients of terms in the expressions of the manifold gradients.

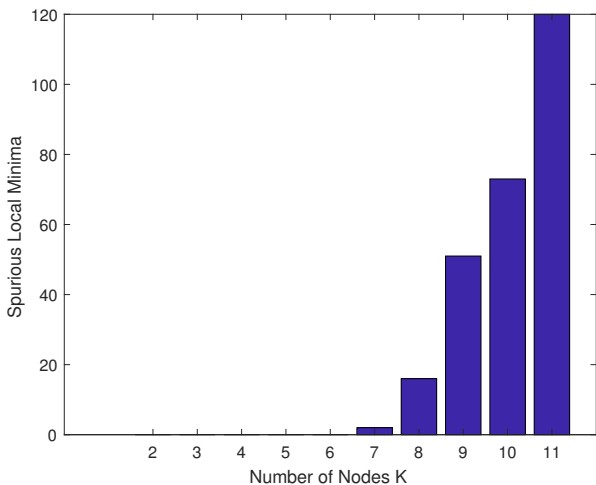

Figure 1: Spurious Local Minima for $K \geq 2$ ReLU Network.

## 7 EXPERIMENTS

We did experiments to verify the theoretical results. Since our results are restricted to the case of $K = 2$ hidden units, it is also natural to investigate whether general two-layer ReLU networks also have the property that all local minima are global minima. Unfortunately as we show via numerical simulation, this is not the case. We consider the cases of $K$ from 2 to 11 hidden units and we set the dimension $d = K$. For each $K$, the true parameters are orthogonal to each other. For each $K$, we run projected gradient descent with 300 different random initializations, and count the number of local minimum (critical points where the manifold Hessian is positive definite) with non-zero training error. If we reach a sub-optimal local minimum, we can conclude the loss surface exhibits spurious local minima. The bar plot showing the number of times gradient descent converged to spurious local minima is in Figure 1. From the plot, we see there is no spurious local minima from $K = 2$ to $K = 6$. However for $K \geq 7$, we observe a clear trend that there are more spurious local minima when there are more hidden units.

## 8 CONCLUSION AND FUTURE WORK

In this paper, we provided recovery guarantee of stochastic gradient descent with random initialization for learning a two-layer neural network with two hidden nodes, unit-norm weights, ReLU activation functions and Gaussian inputs. Experiments are also done to verify our results. For future work, here we list some possible directions.

### 8.1 GENERAL CASE OF NETWORKS

This paper focused on a ReLU network with only two hidden units, . And the teaching weights must be orthogonal. Those are many conditions, in which we think there are some conditions that are not quite essential, e.g., the orthogonal assumption. In experiments we have already seen that even if they are not orthogonal, it still has some good properties such as the positions of critical points. Therefore, in the future we can further relax or abandon some of the assumptions of this paper and preserve or improve the result we have.

### 8.2 BAD LOCAL MINIMA

The neural network we discussed in this paper is in some sense very simple and far from practice, although it is already the most complex model when we want to analyze the whole loss surface. By experiments we have found that when it comes to seven hidden nodes with orthogonal true parameters,

there will be some bad local minima, i.e., there are some local minima that are not global. We believe that research in this paper can capture the characteristics of the whole loss surface and can help analyze the loss surface when there are three or even more hidden units, which may give some bounds on the performance of bad local minima and help us understand the specific non-convexity of loss surfaces.

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

## A    PRELIMINARIES

Consider a neural network with 2 hidden nodes and ReLU as the activation function:

$$F(x) = \frac{\sigma(w_1^T x) + \sigma(w_2^T x)}{2}, \tag{25}$$

where $\sigma(x) = \max(0, x)$ is the ReLU function.

First we study the 2-D case, i.e., the input and all parameters are two dimensional. Assume that the input follows standard normal distribution.

The loss function is population loss:

$$l(W) = \mathbb{E}_x \left[ \left( \frac{\sigma(w_1^T x) + \sigma(w_2^T x)}{2} - \frac{\sigma(w_1^{*T} x) + \sigma(w_2^{*T} x)}{2} \right)^2 \right]. \tag{26}$$

Define

$$g(u, v) = \mathbb{E}_x \left[ \sigma(u^T x) \sigma(v^T x) \right], \tag{27}$$

then from Brutzkus and Globerson (2017) we get

$$g(u, v) = \frac{1}{2\pi} \|u\| \|v\| \left( \sin \theta_{u,v} - (\pi - \theta_{u,v}) \cos \theta_{u,v} \right). \tag{28}$$

Thus,

$$\frac{\partial g}{\partial u} = \frac{1}{2\pi} \|v\| \frac{u}{\|u\|} \sin \theta_{u,v} + \frac{1}{2\pi} (\pi - \theta_{u,v}) v. \tag{29}$$

Moreover, from (26) we get

$$l(W) = \frac{1}{4} \sum_{i,j \in [2]} \left( g(w_i, w_j) - 2g(w_i, w_j^*) + g(w_i^*, w_j^*) \right). \tag{30}$$

Assume $\|w_1^*\| = \|w_2^*\|$ and $w_1^{*T} w_2^* = 0$. WLOG, let $e_1 = w_1^*$ and $e_2 = w_2^*$. Then we know that $\forall i, j \in [2]$, $g(w_i^*, w_j^*)$ is a constant number. Thus, define the objective function(which equals to $4l(W)$ up to an additive constant)

$$f(W) = g(w_1, w_1) + g(w_2, w_2) + 2g(w_1, w_2) - 2 \sum_{i,j \in [2]} g(w_i, w_j^*). \tag{31}$$

Thus,

$$\frac{\partial f}{\partial w_1} = w_1 + \frac{1}{\pi} \|w_2\| \frac{w_1}{\|w_1\|} \sin \theta_{w_1,w_2} + \frac{1}{\pi} (\pi - \theta_{w_1,w_2}) w_2 \tag{32}$$

$$- \frac{1}{\pi} \|w_1^*\| \frac{w_1}{\|w_1\|} \sin \theta_{w_1,w_1^*} - \frac{1}{\pi} (\pi - \theta_{w_1,w_1^*}) w_1^* \tag{33}$$

$$- \frac{1}{\pi} \|w_2^*\| \frac{w_1}{\|w_1\|} \sin \theta_{w_1,w_2^*} - \frac{1}{\pi} (\pi - \theta_{w_1,w_2^*}) w_2^* \tag{34}$$

$$= w_1 + \frac{1}{\pi} \|w_2\| \frac{w_1}{\|w_1\|} \sin \theta_{w_1,w_2} + \frac{1}{\pi} (\pi - \theta_{w_1,w_2}) w_2 \tag{35}$$

$$- \frac{1}{\pi} \frac{w_1}{\|w_1\|} \sin \theta_{w_1,w_1^*} - \frac{1}{\pi} (\pi - \theta_{w_1,w_1^*}) w_1^* \tag{36}$$

$$- \frac{1}{\pi} \frac{w_1}{\|w_1\|} \sin \theta_{w_1,w_2^*} - \frac{1}{\pi} (\pi - \theta_{w_1,w_2^*}) w_2^*. \tag{37}$$

Similarly, for $w_2$, the gradient is

$$\frac{\partial f}{\partial w_2} = w_2 + \frac{1}{\pi} \|w_1\| \frac{w_2}{\|w_2\|} \sin \theta_{w_1,w_2} + \frac{1}{\pi} (\pi - \theta_{w_1,w_2}) w_1 \tag{38}$$

$$- \frac{1}{\pi} \frac{w_2}{\|w_2\|} \sin \theta_{w_2,w_1^*} - \frac{1}{\pi} (\pi - \theta_{w_2,w_1^*}) w_1^* \tag{39}$$

$$- \frac{1}{\pi} \frac{w_2}{\|w_2\|} \sin \theta_{w_2,w_2^*} - \frac{1}{\pi} (\pi - \theta_{w_2,w_2^*}) w_2^*. \tag{40}$$

Assume that $w_1 = (w_{11}, w_{12})$ and $w_2 = (w_{21}, w_{22})$, then the gradient can be expressed in this form:

$$\frac{\partial f}{\partial w_{11}} = w_{11} + \frac{1}{\pi} \frac{\|w_2\|}{\|w_1\|} w_{11} \sin \theta_{w_1,w_2} + \frac{1}{\pi}(\pi - \theta_{w_1,w_2})w_{21} \tag{41}$$

$$- \frac{1}{\pi} \frac{w_{11}}{\|w_1\|} \sin \theta_{w_1,w_1^*} - \frac{1}{\pi}(\pi - \theta_{w_1,w_1^*}) \tag{42}$$

$$- \frac{1}{\pi} \frac{w_{11}}{\|w_1\|} \sin \theta_{w_1,w_2^*} \tag{43}$$

and

$$\frac{\partial f}{\partial w_{12}} = w_{12} + \frac{1}{\pi} \frac{\|w_2\|}{\|w_1\|} w_{12} \sin \theta_{w_1,w_2} + \frac{1}{\pi}(\pi - \theta_{w_1,w_2})w_{22} \tag{44}$$

$$- \frac{1}{\pi} \frac{w_{12}}{\|w_1\|} \sin \theta_{w_1,w_1^*} \tag{45}$$

$$- \frac{1}{\pi} \frac{w_{12}}{\|w_1\|} \sin \theta_{w_1,w_2^*} - \frac{1}{\pi}(\pi - \theta_{w_1,w_2^*}). \tag{46}$$

Because of symmetry, for $w_2$, the gradient is

$$\frac{\partial f}{\partial w_{21}} = w_{21} + \frac{1}{\pi} \frac{\|w_1\|}{\|w_2\|} w_{21} \sin \theta_{w_1,w_2} + \frac{1}{\pi}(\pi - \theta_{w_1,w_2})w_{11} \tag{47}$$

$$- \frac{1}{\pi} \frac{w_{21}}{\|w_2\|} \sin \theta_{w_2,w_1^*} - \frac{1}{\pi}(\pi - \theta_{w_2,w_1^*}) \tag{48}$$

$$- \frac{1}{\pi} \frac{w_{21}}{\|w_2\|} \sin \theta_{w_2,w_2^*} \tag{49}$$

and

$$\frac{\partial f}{\partial w_{22}} = w_{22} + \frac{1}{\pi} \frac{\|w_1\|}{\|w_2\|} w_{22} \sin \theta_{w_1,w_2} + \frac{1}{\pi}(\pi - \theta_{w_1,w_2})w_{12} \tag{50}$$

$$- \frac{1}{\pi} \frac{w_{22}}{\|w_2\|} \sin \theta_{w_2,w_1^*} \tag{51}$$

$$- \frac{1}{\pi} \frac{w_{22}}{\|w_2\|} \sin \theta_{w_2,w_2^*} - \frac{1}{\pi}(\pi - \theta_{w_2,w_2^*}). \tag{52}$$

## B  CRITICAL POINTS IN 2D CASES

### B.1  2D PRELIMINARIES

In 2D cases, we can translate $W$ to polar coordinates and fix $\|w_1\| = \|w_2\| = 1$, so there are two variables left: $\theta_1$ and $\theta_2$, i.e., $w_1 = (\cos \theta_1, \sin \theta_1)$ and $w_2 = (\cos \theta_2, \sin \theta_2)$.

For manifold gradient, we only need to consider its norm and check whether it's zero. For $w_1$ and $w_2$, the (directed) norm of manifold gradients(expressed by $m$) are $m(w_1) = \sin \theta_1 \frac{\partial f}{\partial w_{11}} - \cos \theta_1 \frac{\partial f}{\partial w_{12}}$ and $m(w_2) = \sin \theta_2 \frac{\partial f}{\partial w_{21}} - \cos \theta_2 \frac{\partial f}{\partial w_{22}}$.

To make life easier, it's better to simplify the $m$ functions a bit using $w_1 = (\cos\theta_1, \sin\theta_1)$ and $w_2 = (\cos\theta_2, \sin\theta_2)$:

$$m(w_1) = \sin\theta_1 \frac{\partial f}{\partial w_{11}} - \cos\theta_1 \frac{\partial f}{\partial w_{12}} \tag{53}$$

$$= \sin\theta_1 \left( \cos\theta_1 + \frac{1}{\pi}\cos\theta_1 \sin\theta_{w_1,w_2} + \frac{1}{\pi}(\pi - \theta_{w_1,w_2})\cos\theta_2 \right. \tag{54}$$

$$\left. - \frac{1}{\pi}\cos\theta_1 \sin\theta_{w_1,w_1^*} - 1 + \frac{\theta_{w_1,w_1^*}}{\pi} - \frac{1}{\pi}\cos\theta_1 \sin\theta_{w_1,w_2^*} \right) \tag{55}$$

$$- \cos\theta_1 \left( \sin\theta_1 + \frac{1}{\pi}\sin\theta_1 \sin\theta_{w_1,w_2} + \frac{1}{\pi}(\pi - \theta_{w_1,w_2})\sin\theta_2 \right. \tag{56}$$

$$\left. - \frac{1}{\pi}\sin\theta_1 \sin\theta_{w_1,w_1^*} - 1 + \frac{\theta_{w_1,w_2^*}}{\pi} - \frac{1}{\pi}\sin\theta_1 \sin\theta_{w_1,w_2^*} \right) \tag{57}$$

$$= \frac{1}{\pi}(\pi - \theta_{w_1,w_2})\sin(\theta_1 - \theta_2) + \cos\theta_1 - \sin\theta_1 \tag{58}$$

$$+ \frac{1}{\pi}\left( \theta_{w_1,w_1^*}\sin\theta_1 - \theta_{w_1,w_2^*}\cos\theta_1 \right). \tag{59}$$

Similarly,

$$m(w_2) = \frac{1}{\pi}(\pi - \theta_{w_1,w_2})\sin(\theta_2 - \theta_1) + \cos\theta_2 - \sin\theta_2 \tag{60}$$

$$+ \frac{1}{\pi}\left( \theta_{w_2,w_1^*}\sin\theta_2 - \theta_{w_2,w_2^*}\cos\theta_2 \right). \tag{61}$$

Then we can divide them into several cases and analyze them one by one to specify the positions and properties of the critical points.

WLOG, assume $\theta_1 \le \theta_2$.

## B.2    $0 \le \theta_1 \le \theta_2 \le \frac{\pi}{2}$

The norm of the manifold gradient w.r.t. $w_1$ is

$$m(w_1) = \frac{1}{\pi}(\pi - \theta_2 + \theta_1)\sin(\theta_1 - \theta_2) + \cos\theta_1 - \sin\theta_1 \tag{62}$$

$$+ \frac{1}{\pi}\left( \theta_1 \sin\theta_1 - \left(\frac{\pi}{2} - \theta_1\right)\cos\theta_1 \right). \tag{63}$$

Similarly, the norm of $m(w_2)$ is

$$m(w_2) = \frac{1}{\pi}(\pi - \theta_2 + \theta_1)\sin(\theta_2 - \theta_1) + \cos\theta_2 - \sin\theta_2 \tag{64}$$

$$+ \frac{1}{\pi}\left( \theta_2 \sin\theta_2 - \left(\frac{\pi}{2} - \theta_2\right)\cos\theta_2 \right). \tag{65}$$

Define

$$h_1(\theta) = \cos\theta - \sin\theta + \frac{1}{\pi}\left( \theta\sin\theta - \left(\frac{\pi}{2} - \theta\right)\cos\theta \right). \tag{66}$$

If $m(w_1) = m(w_2) = 0$, then

$$h_1(\theta_1) = \frac{1}{\pi}(\pi - \theta_2 + \theta_1)\sin(\theta_2 - \theta_1) \tag{67}$$

and

$$h_1(\theta_2) = \frac{1}{\pi}(\pi - \theta_2 + \theta_1)\sin(\theta_1 - \theta_2). \tag{68}$$

Thus,

$$h_1(\theta_1) + h_1(\theta_2) = 0. \tag{69}$$

Note that when $0 \le \theta \le \frac{\pi}{2}$,

$$h_1'(\theta) = -\frac{\frac{\pi}{2} - 1 + \theta}{\pi} \sin\theta - \frac{\pi - 1 - \theta}{\pi} \cos\theta < 0. \tag{70}$$

Also note that

$$h_1(\theta) + h_1(\frac{\pi}{2} - \theta) = \cos\theta - \sin\theta + \frac{1}{\pi}\left(\theta\sin\theta - \left(\frac{\pi}{2} - \theta\right)\cos\theta\right) \tag{71}$$

$$+ \cos\left(\frac{\pi}{2} - \theta\right) - \sin\left(\frac{\pi}{2} - \theta\right) \tag{72}$$

$$+ \frac{1}{\pi}\left(\left(\frac{\pi}{2} - \theta\right)\sin\left(\frac{\pi}{2} - \theta\right) - \theta\cos\left(\frac{\pi}{2} - \theta\right)\right) \tag{73}$$

$$= \cos\theta - \sin\theta + \frac{1}{\pi}\left(\theta\sin\theta - \left(\frac{\pi}{2} - \theta\right)\cos\theta\right) \tag{74}$$

$$+ \sin\theta - \cos\theta + \frac{1}{\pi}\left(\left(\frac{\pi}{2} - \theta\right)\cos\theta - \theta\sin\theta\right) \tag{75}$$

$$= 0. \tag{76}$$

Thus, if $m(w_1) = m(w_2) = 0$, then $\theta_1 + \theta_2 = \frac{\pi}{2}$. From $\theta_1 \le \theta_2$ we know that $\theta_1 \le \frac{\pi}{4}$. Plug $\theta_2 = \frac{\pi}{2} - \theta_1$ into (63) and we get

$$m(w_1) = 0 \Leftrightarrow h_1(\theta_1) = \frac{2\theta_1 + \frac{\pi}{2}}{\pi}\cos(2\theta_1). \tag{77}$$

**Lemma B.1.** *If $0 \le \theta \le \frac{\pi}{4}$, then*

$$h_1(\theta) \le \frac{2\theta + \frac{\pi}{2}}{\pi}\cos(2\theta) \tag{78}$$

*and the inequality becomes equality only then $\theta = 0$ or $\theta = \frac{\pi}{4}$.*

*Proof.* When $0 \le \theta \le \frac{\pi}{4}$,

$$\frac{2\theta + \frac{\pi}{2}}{\pi}\cos(2\theta) - h_1(\theta) \tag{79}$$

$$= \left(\frac{1}{2} + \frac{2\theta}{\pi}\right)\cos(2\theta) + \left(1 - \frac{\theta}{\pi}\right)\sin\theta - \left(\frac{1}{2} + \frac{\theta}{\pi}\right)\cos\theta \tag{80}$$

$$\ge \left(\frac{1}{2} + \frac{\theta}{\pi}\right)\cos(2\theta) + \left(1 - \frac{\theta}{\pi}\right)\sin\theta - \left(\frac{1}{2} + \frac{\theta}{\pi}\right)\cos\theta \tag{81}$$

$$\ge \left(\frac{1}{2} + \frac{\theta}{\pi}\right)\cos(2\theta) + \frac{3}{4}\sin\theta - \left(\frac{1}{2} + \frac{\theta}{\pi}\right)\cos\theta \tag{82}$$

$$= \left(\frac{1}{2} + \frac{\theta}{\pi}\right)(\cos(2\theta) - \cos\theta) + \frac{3}{4}\sin\theta \tag{83}$$

$$\ge \frac{3}{4}(\cos(2\theta) - \cos\theta) + \frac{3}{4}\sin\theta \tag{84}$$

$$= \frac{3}{4}(\cos(2\theta) - (\cos\theta - \sin\theta)) \tag{85}$$

$$= \frac{3}{4}\left(\cos^2\theta - \sin^2\theta - (\cos\theta - \sin\theta)\right) \tag{86}$$

$$= \frac{3}{4}(\cos\theta - \sin\theta)(\cos\theta + \sin\theta - 1) \tag{87}$$

$$\ge 0. \tag{88}$$

Note that (84) is because $\cos(2\theta) - \cos\theta$ is always non-positive when $0 \le \theta \le \frac{\pi}{4}$.

From (81), the inequality becomes an equality only when $\theta\cos(2\theta) = 0$, which means that the only possibilities are $\theta = 0$ or $\theta = \frac{\pi}{4}$. After plugging in those two possibilities in (78), we know that $h(\theta) = \frac{2\theta + \frac{\pi}{2}}{\pi}\cos(2\theta)$ holds when $\theta = 0$ or $\theta = \frac{\pi}{4}$. $\qquad\square$

Using the above lemma, we conclude that $m(w_1) = 0$ iff $\theta_1 + \theta_2 = \frac{\pi}{2}$ and $\theta = 0$ or $\frac{\pi}{4}$, i.e., $m(w_1) = 0$ iff $(\theta_1, \theta_2) = (0, \frac{\pi}{2})$ or $(\theta_1, \theta_2) = (\frac{\pi}{4}, \frac{\pi}{4})$.

In a word, there are two critical points in this case: $(\theta_1, \theta_2) = (0, \frac{\pi}{2})$ and $(\theta_1, \theta_2) = (\frac{\pi}{4}, \frac{\pi}{4})$.

### B.3   $\frac{\pi}{2} \leq \theta_1 \leq \theta_2 \leq \pi$

The norm of the manifold gradient w.r.t. $w_1$ is

$$m(w_1) = \frac{1}{\pi}(\pi - \theta_2 + \theta_1)\sin(\theta_1 - \theta_2) + \cos\theta_1 - \sin\theta_1 \tag{89}$$

$$+ \frac{1}{\pi}\left(\theta_1 \sin\theta_1 - \left(\theta_1 - \frac{\pi}{2}\right)\cos\theta_1\right). \tag{90}$$

Similarly,

$$m(w_2) = \frac{1}{\pi}(\pi - \theta_2 + \theta_1)\sin(\theta_2 - \theta_1) + \cos\theta_2 - \sin\theta_2 \tag{91}$$

$$+ \frac{1}{\pi}\left(\theta_2 \sin\theta_2 - \left(\theta_2 - \frac{\pi}{2}\right)\cos\theta_2\right). \tag{92}$$

Define

$$h_2(\theta) = \cos\theta - \sin\theta + \frac{1}{\pi}\left(\theta \sin\theta - \left(\theta - \frac{\pi}{2}\right)\cos\theta\right), \tag{93}$$

Let $\theta' = \theta - \frac{\pi}{2}$, then

$$h_2(\theta) = h_2\left(\theta' + \frac{\pi}{2}\right) \tag{94}$$

$$= -\sin\theta' - \cos\theta' + \frac{1}{\pi}\left(\left(\theta' + \frac{\pi}{2}\right)\sin\left(\theta' + \frac{\pi}{2}\right) - \theta'\cos\left(\theta' + \frac{\pi}{2}\right)\right) \tag{95}$$

$$= -\sin\theta' - \cos\theta' + \frac{1}{\pi}\left(\left(\theta' + \frac{\pi}{2}\right)\cos\theta' + \theta'\sin\theta'\right) \tag{96}$$

$$= -\sin\theta' + \frac{1}{\pi}\left(\left(\theta' - \frac{\pi}{2}\right)\cos\theta' + \theta'\sin\theta'\right) \tag{97}$$

$$= -\sin\theta' + \frac{1}{\pi}\left(\theta'\sin\theta' - \left(\frac{\pi}{2} - \theta'\right)\cos\theta'\right) \tag{98}$$

$$= h_1(\theta') - \cos\theta' \tag{99}$$

$$\tag{100}$$

**Lemma B.2.** *When $\theta \in [\frac{\pi}{2}, \pi]$,*

$$h_2(\theta) \leq -\frac{1}{2}, \tag{101}$$

*and the inequality becomes equality only then $\theta = \frac{\pi}{2}$ or $\theta = \pi$.*

*Proof.* Let $\theta' = \theta - \frac{\pi}{2}$, then $\theta' \in [0, \frac{\pi}{2}]$ and

$$h_2(\theta) = h_1(\theta') - \cos\theta' \tag{102}$$

$$= -\sin\theta' + \frac{1}{\pi}\left(\theta'\sin\theta' - \left(\frac{\pi}{2} - \theta'\right)\cos\theta'\right) \tag{103}$$

$$= \left(\frac{\theta'}{\pi} - \frac{1}{2}\right)\cos\theta' + \left(\frac{\theta'}{\pi} - 1\right)\sin\theta' \tag{104}$$

$$\leq -\frac{1}{2}\cos\theta' - \frac{1}{2}\sin\theta' \tag{105}$$

$$= -\frac{1}{2}(\cos\theta' + \sin\theta') \tag{106}$$

$$\leq -\frac{1}{2}. \tag{107}$$

Note that the inequality becomes equality only when $\theta'\cos\theta' = 0$ and $\left(\frac{\theta'}{\pi} - \frac{1}{2}\right)\sin\theta' = 0$, i.e., $\theta = \frac{\pi}{2}$ or $\theta = \pi$. $\qquad\square$

If $m(w_1) = m(w_2) = 0$, then

$$h_2(\theta_1) = \frac{1}{\pi}(\pi - \theta_2 + \theta_1)\sin(\theta_2 - \theta_1) \tag{108}$$

and

$$h_2(\theta_2) = \frac{1}{\pi}(\pi - \theta_2 + \theta_1)\sin(\theta_1 - \theta_2). \tag{109}$$

Thus,

$$h_2(\theta_1) + h_2(\theta_2) = 0. \tag{110}$$

However, we know that $h_2(\theta_1) < 0$ and $h_2(\theta_1) < 0$, which makes a contradiction.

In a word, there is no critical point in this case.

### B.4  $\pi \le \theta_1 \le \theta_2 \le \frac{3\pi}{2}$

The norm of the manifold gradient w.r.t. $w_1$ is

$$m(w_1) = \frac{1}{\pi}(\pi - \theta_2 + \theta_1)\sin(\theta_1 - \theta_2) + \cos\theta_1 - \sin\theta_1 \tag{111}$$

$$+ \frac{1}{\pi}\left((2\pi - \theta_1)\sin\theta_1 - \left(\theta_1 - \frac{\pi}{2}\right)\cos\theta_1\right). \tag{112}$$

Similarly, the norm of $m(w_2)$ is

$$m(w_2) = \frac{1}{\pi}(\pi - \theta_2 + \theta_1)\sin(\theta_2 - \theta_1) + \cos\theta_2 - \sin\theta_2 \tag{113}$$

$$+ \frac{1}{\pi}\left((2\pi - \theta_2)\sin\theta_2 - \left(\theta_2 - \frac{\pi}{2}\right)\cos\theta_2\right). \tag{114}$$

Define

$$h_3(\theta) = \cos\theta - \sin\theta + \frac{1}{\pi}\left((2\pi - \theta)\sin\theta - \left(\theta - \frac{\pi}{2}\right)\cos\theta\right). \tag{115}$$

Let $\theta = \theta' + \pi$, then

$$h_3(\theta) = h_3(\theta' + \pi) \tag{116}$$

$$= \cos(\theta' + \pi) - \sin(\theta' + \pi) \tag{117}$$

$$+ \frac{1}{\pi}\left((\pi - \theta')\sin(\theta' + \pi) - \left(\theta' + \frac{\pi}{2}\right)\cos(\theta' + \pi)\right) \tag{118}$$

$$= -\cos\theta' + \sin\theta' + \frac{1}{\pi}\left((\pi - \theta')(-\sin\theta') - \left(\pi + \theta' - \frac{\pi}{2}\right)(-\cos\theta')\right) \tag{119}$$

$$= -\cos\theta' + \sin\theta' + \frac{1}{\pi}\left(-\pi\sin\theta' + \theta'\sin\theta' + \pi\cos\theta' + \left(\theta' - \frac{\pi}{2}\right)\cos\theta'\right) \tag{120}$$

$$= -\cos\theta' + \sin\theta' - \sin\theta' + \cos\theta' + \frac{1}{\pi}\left(\theta'\sin\theta' - \left(\frac{\pi}{2} - \theta'\right)\cos\theta'\right) \tag{121}$$

$$= \frac{1}{\pi}\left(\theta'\sin\theta' - \left(\frac{\pi}{2} - \theta'\right)\cos\theta'\right) \tag{122}$$

$$= h_1(\theta') - \cos\theta' + \sin\theta'. \tag{123}$$

Moreover, $\forall\theta \in [\pi, \frac{3\pi}{2}]$,

$$h_3(\theta) + h_3(\frac{5\pi}{2} - \theta) = h_1(\theta - \pi) - \cos(\theta - \pi) + \sin(\theta - \pi) \tag{124}$$

$$+ h_1(\frac{5\pi}{2} - \theta - \pi) - \cos(\frac{5\pi}{2} - \theta - \pi) + \sin(\frac{5\pi}{2} - \theta - \pi) \tag{125}$$

$$= h_1(\theta - \pi) + \cos\theta - \sin\theta + h_1(\frac{3\pi}{2} - \theta) + \sin\theta - \cos\theta \tag{126}$$

$$= h_1(\theta - \pi) + h_1(\frac{3\pi}{2} - \theta) \tag{127}$$

$$= 0. \tag{128}$$

Also, when $\theta \in [\pi, \frac{3\pi}{2}]$,

$$h_3'(\theta) = \frac{\pi - \theta - 1}{\pi} \cos\theta + \frac{\theta - \frac{3\pi}{2} - 1}{\pi} \sin\theta > 0, \tag{129}$$

so $h_3$ is an increasing function when $\theta \in [\pi, \frac{3\pi}{2}]$.

Thus, if $m(w_1) = m(w_2) = 0$, then $\theta_1 + \theta_2 = \frac{5\pi}{2}$. From $\theta_1 \leq \theta_2$ we know that $\theta_1 \leq \frac{5\pi}{4}$. Plug $\theta_2 = \frac{5\pi}{2} - \theta_1$ in (112) and we get

$$m(w_1) = 0 \Leftrightarrow h_3(\theta_1) = \frac{2\theta_1 - \frac{3\pi}{2}}{\pi} \cos(2\theta_1). \tag{130}$$

From Lemma B.1,

$$h_3(\theta_1) = h_1(\theta_1 - \pi) - \cos(\theta_1 - \pi) + \sin(\theta_1 - \pi) \tag{131}$$

$$\leq \frac{2\theta_1 - \frac{3\pi}{2}}{\pi} \cos(2(\theta_1) - \pi) - \cos(\theta_1 - \pi) + \sin(\theta_1 - \pi) \tag{132}$$

$$= \frac{2\theta_1 - \frac{3\pi}{2}}{\pi} \cos(2\theta_1) - \cos(\theta_1 - \pi) + \sin(\theta_1 - \pi) \tag{133}$$

$$\leq \frac{2\theta_1 - \frac{3\pi}{2}}{\pi} \cos(2\theta_1). \tag{134}$$

Note that (132) becomes equality only when $\theta_1 = \pi$ or $\theta_1 = \frac{5\pi}{4}$, and (134) becomes equality only when $\theta_1 = \frac{5\pi}{4}$. Therefore, in this case, $m(w_1) = 0$ if and only if $\theta_1 = \frac{5\pi}{4}$.

In a word, the only critical point in this case is $(\theta_1, \theta_2) = (\frac{5\pi}{4}, \frac{5\pi}{4})$.

## B.5 $\frac{3\pi}{2} \leq \theta_1 \leq \theta_2 \leq 2\pi$

Actually, this is symmetric to the B.3, so in this part I would like to specify this kind of symmetry.

We have already assumed that $\theta_1 \leq \theta_2$ without loss of generality, and under this assumption, we can find another symmetry: From $w_1$ and $w_2$, using line $y = x$ as symmetry axis, we can get two new vectors $w_1'$ and $w_2'$. $w_1'$ is not necessarily the image of $w_1$ because we need to preserve the assumption that $\theta_1 \leq \theta_2$, but there exists one and only one mapping such that $\theta_1' \leq \theta_2'$. In this kind of symmetry, the angles, including $\theta_{w_1, w_2}$ and $\theta_{w_i, w_j^*}$ where $i, j \in [2]$, are the same, so the two symmetric cases share the same gradients, thus the symmetric critical points.

We use $(i, j)$, where $i, j \in [4]$, to represent the case that $\theta_1$ is in the $i$th quadrant and $\theta_2$ is in the $j$th one. Using this kind of symmetry, we conclude that $(1, 2)$ is equivalent to $(1, 4)$ and $(2, 3)$ is equivalent to $(3, 4)$, so there are 4 cases left which are $(1, 2)$, $(1, 3)$, $(2, 3)$ and $(2, 4)$.

## B.6 $0 \leq \theta_1 \leq \frac{\pi}{2} \leq \theta_2 \leq \pi$

Similar to previous cases,

$$m(w_1) = \frac{1}{\pi}(\pi - \theta_2 + \theta_1) \sin(\theta_1 - \theta_2) + \cos\theta_1 - \sin\theta_1 \tag{135}$$

$$+ \frac{1}{\pi}\left(\theta_1 \sin\theta_1 - \left(\frac{\pi}{2} - \theta_1\right)\cos\theta_1\right) \tag{136}$$

and

$$m(w_2) = \frac{1}{\pi}(\pi - \theta_2 + \theta_1) \sin(\theta_2 - \theta_1) + \cos\theta_2 - \sin\theta_2 \tag{137}$$

$$+ \frac{1}{\pi}\left(\theta_2 \sin\theta_2 - \left(\theta_2 - \frac{\pi}{2}\right)\cos\theta_2\right). \tag{138}$$

Using previous definitions, we conclude that

$$m(w_1) = \frac{1}{\pi}(\pi - \theta_2 + \theta_1) \sin(\theta_1 - \theta_2) + h_1(\theta_1) \tag{139}$$

and

$$m(w_2) = \frac{1}{\pi}(\pi - \theta_2 + \theta_1)\sin(\theta_2 - \theta_1) + h_2(\theta_2). \tag{140}$$

If $m(w_1) = m(w_2) = 0$, then $m_{(}w_1) + m(w_2) = 0$, i.e.,

$$h_1(\theta_1) + h_2(\theta_2) = 0. \tag{141}$$

From (99) we know that

$$h_1(\theta_1) = h_2(\theta_1 + \frac{\pi}{2}) + \cos\theta_1. \tag{142}$$

Thus, using lemma B.2,

$$h_1(\theta_1) + h_2(\theta_2) = h_2(\theta_1 + \frac{\pi}{2}) + h_2(\theta_2) + \cos\theta_1 \leq -\frac{1}{2} - \frac{1}{2} + 1 = 0. \tag{143}$$

That means the only case that $h_1(\theta_1) + h_2(\theta_2) = 0$ is when the inequality (143) becomes equality, which means that $\cos\theta_1 = 1$ and $h_2(\theta_1 + \frac{\pi}{2}) = h_2(\theta_2) = -\frac{1}{2}$. Thus, we must have $\theta_1 = 0$, and $\theta_2 = \frac{\pi}{2}$ or $\theta_2 = \pi$. Plugging them back in (136) and (138), we can verify that the first one is a critical point while the other is not. Since $(\theta_1, \theta_2) = (0, \frac{\pi}{2})$ has been counted in case 1, there are no new critical points in this case.

## B.7 $\quad 0 \leq \theta_1 \leq \frac{\pi}{2}, \pi \leq \theta_2 \leq \frac{3\pi}{2}$

Similar to previous cases,

$$m(w_1) = \frac{1}{\pi}(\pi - \theta_{w_1,w_2})\sin(\theta_1 - \theta_2) + \cos\theta_1 - \sin\theta_1 \tag{144}$$

$$+ \frac{1}{\pi}\left(\theta_1\sin\theta_1 - \left(\frac{\pi}{2} - \theta_1\right)\cos\theta_1\right) \tag{145}$$

and

$$m(w_2) = \frac{1}{\pi}(\pi - \theta_{w_1,w_2})\sin(\theta_2 - \theta_1) + \cos\theta_2 - \sin\theta_2 \tag{146}$$

$$+ \frac{1}{\pi}\left((2\pi - \theta_2)\sin\theta_2 - \left(\theta_2 - \frac{\pi}{2}\right)\cos\theta_2\right). \tag{147}$$

Thus, using previous definitions

$$m(w_1) = \frac{1}{\pi}(\pi - \theta_{w_1,w_2})\sin(\theta_1 - \theta_2) + h_1(\theta_1) \tag{148}$$

and

$$m(w_2) = \frac{1}{\pi}(\pi - \theta_{w_1,w_2})\sin(\theta_2 - \theta_1) + h_3(\theta_2). \tag{149}$$

If $m(w_1) = m(w_2) = 0$, then $m(w_1) + m(w_2) = 0$, i.e.,

$$h_1(\theta_1) + h_3(\theta_2) = 0. \tag{150}$$

For $0 \leq \theta \leq \frac{\pi}{2}$, define

$$H(\theta) = h_1(\theta) + h_3(\theta + \pi). \tag{151}$$

Then we have the following lemma:

**Lemma B.3.** *When $0 \leq \theta \leq \frac{\pi}{4}$, $H(\theta) \leq 0$, and when $\frac{\pi}{4} \leq \theta \leq \frac{\pi}{2}$, $H(\theta) \geq 0$. Besides, all zero points of $H$ in $[0, \frac{\pi}{2}]$ are $\theta = 0, \frac{\pi}{4}$ and $\frac{\pi}{2}$.*

*Proof.* From (123), $h_3(\theta + \pi) = h_1(\theta) - \cos\theta + \sin\theta$. Thus,

$$H(\theta) = 2h_1(\theta) - \cos\theta + \sin\theta \tag{152}$$

$$= \cos\theta - \sin\theta + \frac{2}{\pi}\left(\theta\sin\theta - \left(\frac{\pi}{2} - \theta\right)\cos\theta\right) \tag{153}$$

$$= \frac{2\theta}{\pi}\cos\theta + \left(\frac{2\theta}{\pi} - 1\right)\sin\theta \tag{154}$$

$$= \frac{2\theta}{\pi}(\cos\theta + \sin\theta) - \sin\theta. \tag{155}$$

When $0 \le \theta \le \frac{\pi}{4}$, since $\sin \theta$ is a concave function for $\theta$, we know that

$$\sin \theta \ge \frac{\sin \frac{\pi}{4}}{\frac{\pi}{4}} \theta = \frac{2\sqrt{2}}{\pi} \theta. \tag{156}$$

Thus,

$$H(\theta) = \frac{2\theta}{\pi}(\cos \theta + \sin \theta) - \sin \theta \tag{157}$$

$$\le \frac{2\sqrt{2}}{\pi}\theta - \sin \theta \tag{158}$$

$$\le 0. \tag{159}$$

To make $H(\theta) = 0$, we must have $\sin \theta = \frac{2\sqrt{2}}{\pi}\theta$, so $\theta = 0$ or $\theta = \frac{\pi}{4}$.

Besides, when $\frac{\pi}{4} < \theta \le \frac{\pi}{2}$, note that

$$H(\frac{\pi}{2} - \theta) + H(\theta) = 2h_1(\theta) - \cos \theta + \sin \theta \tag{160}$$

$$+ 2h_1(\frac{\pi}{2} - \theta) - \cos(\frac{\pi}{2} - \theta) + \sin(\frac{\pi}{2} - \theta) \tag{161}$$

$$= 2\left(h_1(\theta) + h_1(\frac{\pi}{2} - \theta)\right) \tag{162}$$

$$- \cos \theta + \sin \theta - \cos(\frac{\pi}{2} - \theta) + \sin(\frac{\pi}{2} - \theta) \tag{163}$$

$$= 0. \tag{164}$$

Thus, $H(\theta) = -H(\frac{\pi}{2} - \theta) \ge 0$. And to make $H(\theta) = 0$, the only possibility is $\theta = \frac{\pi}{2}$, which ends the proof. $\square$

Remember that if $m(w_1) = m(w_2) = 0$, then we have $h_3(\theta_2) = -h_1(\theta_1)$.

If $h_1(\theta_1) > 0$, i.e., $0 \le \theta_1 < \frac{\pi}{4}$, then from lemma B.3, $H(\theta_1) \le 0$, which means that

$$h_3(\theta_1 + \pi) \le -h_1(\theta_1). \tag{165}$$

Since $h_3$ is a strictly increasing function, we know that if $h_3(\theta_2) = -h_1(\theta_1)$, then $\theta_2 \ge \theta_1 + \pi$, so $\sin(\theta_1 - \theta_2) \ge 0$, and that means

$$m(w_1) = \frac{1}{\pi}(\pi - \theta_{w_1,w_2})\sin(\theta_1 - \theta_2) + h_1(\theta_1) > 0 + 0 = 0. \tag{166}$$

Similarly, if $h_1(\theta_1) < 0$, i.e., $\frac{\pi}{4} < \theta_1 \le \frac{\pi}{2}$, then from lemma B.3, $H(\theta_1) \ge 0$, which means that

$$h_3(\theta_1 + \pi) \ge -h_1(\theta_1). \tag{167}$$

Thus, if $h_3(\theta_2) = -h_1(\theta_1)$, then $\theta_2 \le \theta_1 + \pi$, so $\sin(\theta_1 - \theta_2) \le 0$, and that means

$$m(w_1) = \frac{1}{\pi}(\pi - \theta_{w_1,w_2})\sin(\theta_1 - \theta_2) + h_1(\theta_1) < 0 + 0 = 0. \tag{168}$$

The last possibility is $h_1(\theta_1) = 0$, i.e., $\theta_1 = \frac{\pi}{4}$. Plugging it into (150) and we know that $h_3(\theta_2) = 0$, so $\theta_2 = \frac{5\pi}{4}$. And that is indeed a critical point.

In a word, the only critical point in this case is $(\theta_1, \theta_2) = (\frac{\pi}{4}, \frac{5\pi}{4})$.

### B.8 $\quad \frac{\pi}{2} \le \theta_1 \le \pi \le \theta_2 \le \frac{3\pi}{2}$

Like previous cases,

$$m(w_1) = \frac{1}{\pi}(\pi - \theta_{w_1,w_2})\sin(\theta_1 - \theta_2) + h_2(\theta_1) \tag{169}$$

and

$$m(w_2) = \frac{1}{\pi}(\pi - \theta_{w_1,w_2})\sin(\theta_2 - \theta_1) + h_3(\theta_2). \tag{170}$$

If $m(w_1) = m(w_2) = 0$, then $m_{(}w_1) + m(w_2) = 0$, i.e.,

$$h_2(\theta_1) + h_3(\theta_2) = 0. \tag{171}$$

Let $\theta' = \theta_2 - \pi$, then from (99) and (123), we know that

$$h_3(\theta_2) = h_3(\theta' + \pi) \tag{172}$$

$$= h_1(\theta') - \cos\theta' + \sin\theta' \tag{173}$$

$$= h_2(\theta' + \frac{\pi}{2}) + \sin\theta'. \tag{174}$$

Thus, from lemma B.2,

$$h_2(\theta_1) + h_3(\theta_2) = h_2(\theta_1) + h_2(\theta_2 - \frac{\pi}{2}) + \sin(\theta_2 - \pi) \tag{175}$$

$$\leq -\frac{1}{2} - \frac{1}{2} + 1 \tag{176}$$

$$= 0. \tag{177}$$

Therefore, in order to achieve $h_2(\theta_1) + h_3(\theta_2) = 0$, the only way is let (176) becomes equality, which means that $\theta_2 = \frac{3\pi}{2}$ and $\theta_1 = \frac{\pi}{2}$ or $\pi$. Plugging them into (169) and (170) we conclude that both of them are not critical points.

In a word, there is no critical point in this case.

## B.9 $\quad \frac{\pi}{2} \leq \theta_1 \leq \pi, \frac{3\pi}{2} \leq \theta_2 < 2\pi$

Similar to previous cases,

$$m(w_1) = \frac{1}{\pi}(\pi - \theta_{w_1,w_2})\sin(\theta_1 - \theta_2) + h_2(\theta_1) \tag{178}$$

and

$$m(w_2) = \frac{1}{\pi}(\pi - \theta_{w_1,w_2})\sin(\theta_2 - \theta_1) + \cos\theta_2 - \sin\theta_2 \tag{179}$$

$$+ \frac{1}{\pi}\left((2\pi - \theta_2)\sin\theta_2 - \left(\frac{5\pi}{2} - \theta_2\right)\cos\theta_2\right). \tag{180}$$

From $\frac{\pi}{2} \leq \theta_1 \leq \pi$ and $\frac{3\pi}{2} \leq \theta_2 \leq 2\pi$ we know that $\theta_{w_1,w_2} \geq \frac{\pi}{2}$, so

$$\left|\frac{1}{\pi}(\pi - \theta_{w_1,w_2})\sin(\theta_1 - \theta_2)\right| \leq \frac{\frac{\pi}{2}}{\pi} \cdot 1 = \frac{1}{2}. \tag{181}$$

When $\left|\frac{1}{\pi}(\pi - \theta_{w_1,w_2})\sin(\theta_1 - \theta_2)\right| = \frac{1}{2}$, we must have $\theta_{w_1,w_2} = \frac{\pi}{2}$, so it must be true that $(\theta_1, \theta_2) = (\pi, \frac{3\pi}{2})$. However, when $(\theta_1, \theta_2) = (\pi, \frac{3\pi}{2})$, we have $\frac{1}{\pi}(\pi - \theta_{w_1,w_2})\sin(\theta_1 - \theta_2) = -\frac{1}{2}$. Thus,

$$\frac{1}{\pi}(\pi - \theta_{w_1,w_2})\sin(\theta_1 - \theta_2) < \frac{1}{2}. \tag{182}$$

Therefore, using lemma B.2,

$$m(w_1) < \frac{1}{2} + (-\frac{1}{2}) = 0. \tag{183}$$

In a word, there is no critical point in this case.

## B.10 CONCLUSION

In conclusion, based on the assumption that $\theta_1 \leq \theta_2$ there are four critical points in the 2D case: $(\theta_1, \theta_2) = (0, \frac{\pi}{2}), (\frac{\pi}{4}, \frac{\pi}{4}), (\frac{\pi}{4}, \frac{5\pi}{4})$ and $(\frac{5\pi}{4}, \frac{5\pi}{4})$.

## C    HESSIAN FOR 2D CASES

There are 4 critical points: $(\frac{\pi}{4}, \frac{\pi}{4})$, $(\frac{\pi}{4}, \frac{5\pi}{4})$, $(\frac{5\pi}{4}, \frac{5\pi}{4})$, $(0, \frac{\pi}{2})$. Obviously, the point $(0, \frac{\pi}{2})$ is a global minima. Next we want to compute the Hessian on other 3 points.

Assume the manifold is $\mathcal{R} = \{(w_1, w_2) : \|w_1\|_2 = \|w_2\|_2 = 1\}$, then the Hessian on the manifold is

$$z^T \nabla_{\mathcal{R}}^2 f z = z^T \nabla^2 f z - (w_1^T \frac{\partial f}{\partial w_1}) \|z_1\|^2 - (w_2^T \frac{\partial f}{\partial w_2}) \|z_2\|^2 \tag{184}$$

$$= z_1^T \frac{\partial^2 f}{\partial w_1 \partial w_1^T} z_1 + z_2^T \frac{\partial^2 f}{\partial w_2 \partial w_2^T} z_2 + 2z_1^T \frac{\partial^2 f}{\partial w_1 \partial w_2^T} z_2 \tag{185}$$

$$- (w_1^T \frac{\partial f}{\partial w_1}) \|z_1\|^2 - (w_2^T \frac{\partial f}{\partial w_2}) \|z_2\|^2 \tag{186}$$

where $z = (z_1, z_2)$ satisfies $w_1^T z_1 = 0$, $w_2^T z_2 = 0$.

Next, we compute each term in Hessian.

Since

$$\frac{\partial f}{\partial w_1} = w_1 + \frac{1}{\pi} \|w_2\| \frac{w_1}{\|w_1\|} \sin \theta_{w_1, w_2} + \frac{1}{\pi} (\pi - \theta_{w_1, w_2}) w_2 \tag{187}$$

$$- \frac{1}{\pi} \frac{w_1}{\|w_1\|} \sin \theta_{w_1, w_1^*} - \frac{1}{\pi} (\pi - \theta_{w_1, w_1^*}) w_1^* \tag{188}$$

$$- \frac{1}{\pi} \frac{w_1}{\|w_1\|} \sin \theta_{w_1, w_2^*} - \frac{1}{\pi} (\pi - \theta_{w_1, w_2^*}) w_2^*. \tag{189}$$

and

$$\frac{\partial f}{\partial w_2} = w_2 + \frac{1}{\pi} \|w_1\| \frac{w_2}{\|w_2\|} \sin \theta_{w_1, w_2} + \frac{1}{\pi} (\pi - \theta_{w_1, w_2}) w_1 \tag{190}$$

$$- \frac{1}{\pi} \frac{w_2}{\|w_2\|} \sin \theta_{w_2, w_1^*} - \frac{1}{\pi} (\pi - \theta_{w_2, w_1^*}) w_1^* \tag{191}$$

$$- \frac{1}{\pi} \frac{w_2}{\|w_2\|} \sin \theta_{w_2, w_2^*} - \frac{1}{\pi} (\pi - \theta_{w_2, w_2^*}) w_2^*. \tag{192}$$

Then we can get when $w_1 \neq w_2$ and $w_1 \neq -w_2$,

$$
\begin{aligned}
\frac{\partial^2 f}{\partial w_1 \partial w_1^T} = I &+ \frac{\|w_2\|}{\pi} \Big( \frac{\sin \theta_{w_1,w_2}}{\|w_1\|} I - \frac{\sin \theta_{w_1,w_2}}{\|w_1\|^3} w_1 w_1^T \\
&- \frac{\cos \theta_{w_1,w_2}}{\|w_1\|} \frac{1}{\sqrt{1 - (\frac{w_1^T w_2}{\|w_1\| \|w_2\|})^2}} \Big( \frac{w_1 w_2^T}{\|w_1\| \|w_2\|} - \frac{w_1^T w_2}{\|w_1\|^3 \|w_2\|} w_1 w_1^T \Big) \Big) \\
&+ \frac{1}{\pi \sqrt{1 - (\frac{w_1^T w_2}{\|w_1\| \|w_2\|})^2}} \Big( \frac{w_2 w_2^T}{\|w_1\| \|w_2\|} - \frac{w_1^T w_2}{\|w_1\|^3 \|w_2\|} w_2 w_1^T \Big) \\
&- \frac{1}{\pi} \Big( \frac{\sin \theta_{w_1,w_1^*}}{\|w_1\|} I - \frac{\sin \theta_{w_1,w_1^*}}{\|w_1\|^3} w_1 w_1^T \\
&- \frac{\cos \theta_{w_1,w_1^*}}{\|w_1\|} \frac{1}{\sqrt{1 - (\frac{w_1^T w_1^*}{\|w_1\| \|w_1^*\|})^2}} \Big( \frac{w_1 w_1^{*T}}{\|w_1\| \|w_1^*\|} - \frac{w_1^T w_1^*}{\|w_1\|^3 \|w_1^*\|} w_1 w_1^T \Big) \Big) \\
&- \frac{1}{\pi \sqrt{1 - (\frac{w_1^T w_1^*}{\|w_1\| \|w_1^*\|})^2}} \Big( \frac{w_1^* w_1^{*T}}{\|w_1\| \|w_1^*\|} - \frac{w_1^T w_1^*}{\|w_1\|^3 \|w_1^*\|} w_1^* w_1^T \Big) \\
&- \frac{1}{\pi} \Big( \frac{\sin \theta_{w_1,w_2^*}}{\|w_1\|} I - \frac{\sin \theta_{w_1,w_2^*}}{\|w_1\|^3} w_1 w_1^T \\
&- \frac{\cos \theta_{w_1,w_2^*}}{\|w_1\|} \frac{1}{\sqrt{1 - (\frac{w_1^T w_2^*}{\|w_1\| \|w_2^*\|})^2}} \Big( \frac{w_1 w_2^{*T}}{\|w_1\| \|w_2^*\|} - \frac{w_1^T w_2^*}{\|w_1\|^3 \|w_2^*\|} w_1 w_1^T \Big) \Big) \\
&- \frac{1}{\pi \sqrt{1 - (\frac{w_1^T w_2^*}{\|w_1\| \|w_2^*\|})^2}} \Big( \frac{w_2^* w_2^{*T}}{\|w_1\| \|w_2^*\|} - \frac{w_1^T w_2^*}{\|w_1\|^3 \|w_2^*\|} w_2^* w_1^T \Big)
\end{aligned}
$$

Using the fact that $w_1^T z_1 = 0$,

$$
\begin{aligned}
z_1^T \frac{\partial^2 f}{\partial w_1 \partial w_1^T} z_1 = 1 &+ \frac{\sin \theta_{w_1,w_2}}{\pi} + \frac{1}{\pi \sqrt{1 - (w_1^T w_2)^2}} (z_1^T w_2)^2 \\
&- \frac{\sin \theta_{w_1,w_1^*}}{\pi} - \frac{1}{\pi \sqrt{1 - (w_1^T w_1^*)^2}} (z_1^T w_1^*)^2 \\
&- \frac{\sin \theta_{w_1,w_2^*}}{\pi} - \frac{1}{\pi \sqrt{1 - (w_1^T w_2^*)^2}} (z_1^T w_2^*)^2
\end{aligned}
$$

Similarly,

$$
\begin{aligned}
z_2^T \frac{\partial^2 f}{\partial w_2 \partial w_2^T} z_2 = 1 &+ \frac{\sin \theta_{w_1,w_2}}{\pi} + \frac{1}{\pi \sqrt{1 - (w_1^T w_2)^2}} (z_2^T w_1)^2 \\
&- \frac{\sin \theta_{w_2,w_1^*}}{\pi} - \frac{1}{\pi \sqrt{1 - (w_2^T w_1^*)^2}} (z_2^T w_1^*)^2 \\
&- \frac{\sin \theta_{w_2,w_2^*}}{\pi} - \frac{1}{\pi \sqrt{1 - (w_2^T w_2^*)^2}} (z_2^T w_2^*)^2
\end{aligned}
$$

Next,

$$
\frac{\partial^2 f}{\partial w_1 \partial w_2^T} = \frac{\sin \theta_{w_1,w_2}}{\pi \|w_1\| \|w_2\|} w_1 w_2^T
$$
$$
- \frac{\|w_2\| \cos \theta_{w_1,w_2}}{\pi \|w_1\|} \frac{1}{\sqrt{1 - (\frac{w_1^T w_2}{\|w_1\|\|w_2\|})^2}} \left( \frac{1}{\|w_1\| \|w_2\|} w_1 w_1^T - \frac{w_1^T w_2}{\|w_1\| \|w_2\|^3} w_1 w_2^T \right)
$$
$$
+ \frac{1}{\pi \sqrt{1 - (\frac{w_1^T w_2}{\|w_1\|\|w_2\|})^2}} \left( \frac{1}{\|w_1\| \|w_2\|} w_2 w_1^T - \frac{w_1^T w_2}{\|w_1\| \|w_2\|^3} w_2 w_2^T \right) + \frac{1}{\pi}(\pi - \theta_{w_1,w_2})I
$$

and

$$
z_1^T \frac{\partial^2 f}{\partial w_1 \partial w_2^T} z_2 = \frac{1}{\pi \sqrt{1 - (w_1^T w_2)^2}} z_1^T w_2 w_1^T z_2 + \frac{1}{\pi}(\pi - \theta_{w_1,w_2}) z_1^T z_2
$$

In conclusion,

$$
z^T \nabla_R^2 f z =
$$
$$
\left( \frac{1}{\pi \sqrt{1 - (w_1^T w_2)^2}} (z_1^T w_2)^2 - \frac{1}{\pi \sqrt{1 - (w_1^T w_1^*)^2}} (z_1^T w_1^*)^2 - \frac{1}{\pi \sqrt{1 - (w_1^T w_2^*)^2}} (z_1^T w_2^*)^2 \right)
$$
$$
+ \frac{1}{\pi \sqrt{1 - (w_1^T w_2)^2}} (z_2^T w_1)^2 - \frac{1}{\pi \sqrt{1 - (w_2^T w_1^*)^2}} (z_2^T w_1^*)^2 - \frac{1}{\pi \sqrt{1 - (w_2^T w_2^*)^2}} (z_2^T w_2^*)^2
$$
$$
+ \left( \frac{2}{\pi \sqrt{1 - (w_1^T w_2)^2}} z_1^T w_2 w_1^T z_2 + \frac{2}{\pi}(\pi - \theta_{w_1,w_2}) z_1^T z_2 \right)
$$
$$
- \left( \frac{1}{\pi}(\pi - \theta_{w_1,w_2}) w_1^T w_2 - \frac{1}{\pi}(\pi - \theta_{w_1,w_1^*}) w_1^T w_1^* - \frac{1}{\pi}(\pi - \theta_{w_1,w_2^*}) w_1^T w_2^* \right)
$$
$$
- \left( \frac{1}{\pi}(\pi - \theta_{w_1,w_2}) w_2^T w_1 - \frac{1}{\pi}(\pi - \theta_{w_2,w_1^*}) w_2^T w_1^* - \frac{1}{\pi}(\pi - \theta_{w_2,w_2^*}) w_2^T w_2^* \right).
$$

When $w_1 = w_2$ or $w_1 = -w_2$, we should consider the limit of the Hessian.

First, let's compute the limit of some functions that we will use later. For simplicity, we just consider the case when $w_1 \to w_2$. The case $w_1 \to -w_2$ will be the same.

Claim: $\lim_{w_2 \to w_1} \frac{(z_1^T w_2)^2}{\sqrt{1 - (w_1^T w_2)^2}} = 0$

Proof: WLOG, we assume $w_1 = (1, 0)$, $w_2 = (\cos \theta, \sin \theta)$, $\theta \to 0$. Otherwise, we can do a rotation which doesn't affect the inner product. Since $z_1^T w_1 = 0$, $z_1 = (0, 1)$. Then

$$
\lim_{w_2 \to w_1} \frac{(z_1^T w_2)^2}{\sqrt{1 - (w_1^T w_2)^2}} = \lim_{\theta \to 0} \frac{\sin^2 \theta}{\sqrt{1 - \cos^2 \theta}}
$$
$$
= \lim_{\theta \to 0} |\sin \theta|
$$
$$
= 0
$$

∎

Similarly, we have the following claims.

Claim: $\lim_{w_2 \to w_1} \frac{(z_2^T w_1)^2}{\sqrt{1 - (w_1^T w_2)^2}} = 0$

Claim: $\lim\limits_{w_2 \to w_1} \dfrac{z_1^T w_2 w_1^T z_2}{\sqrt{1 - \left(w_1^T w_2\right)^2}} = 0$

Using these claims, we can computet the Hessian when $w_1 = w_2$.

$$\lim_{w_2 \to w_1} z_1^T \frac{\partial^2 f}{\partial w_1 \partial w_1^T} z_1 = 1 - \frac{\sin \theta_{w_1, w_1^*}}{\pi} - \frac{1}{\pi \sqrt{1 - \left(w_1^T w_1^*\right)^2}} \left(z_1^T w_1^*\right)^2$$
$$- \frac{\sin \theta_{w_1, w_2^*}}{\pi} - \frac{1}{\pi \sqrt{1 - \left(w_1^T w_2^*\right)^2}} \left(z_1^T w_2^*\right)^2$$

$$\lim_{w_2 \to w_1} z_2^T \frac{\partial^2 f}{\partial w_2 \partial w_2^T} z_2 = 1 - \frac{\sin \theta_{w_1, w_1^*}}{\pi} - \frac{1}{\pi \sqrt{1 - \left(w_1^T w_1^*\right)^2}} \left(z_1^T w_1^*\right)^2$$
$$- \frac{\sin \theta_{w_1, w_2^*}}{\pi} - \frac{1}{\pi \sqrt{1 - \left(w_1^T w_2^*\right)^2}} \left(z_1^T w_2^*\right)^2$$

$$\lim_{w_2 \to w_1} z_1^T \frac{\partial^2 f}{\partial w_1 \partial w_2^T} z_2 = z_1^T z_2$$

Now, we can compute the hessian on critical points. For simplicity we just consider the case that $k = 1$.

## C.1 $\left(\frac{\pi}{4}, \frac{\pi}{4}\right)$

On the direction $z = (z_1, z_2) = \left(\frac{\sqrt{2}}{2}, -\frac{\sqrt{2}}{2}, \frac{\sqrt{2}}{2}, -\frac{\sqrt{2}}{2}\right)$,

$$w_1^T \frac{\partial f}{\partial w_1} = 2 - \frac{\sqrt{2}}{\pi} - \frac{3\sqrt{2}}{4}$$
$$w_2^T \frac{\partial f}{\partial w_2} = 2 - \frac{\sqrt{2}}{\pi} - \frac{3\sqrt{2}}{4}$$

So

$$z^T \nabla_R^2 f z = z^T \nabla^2 f z - (w_1^T \frac{\partial f}{\partial w_1}) \|z_1\|^2 - (w_2^T \frac{\partial f}{\partial w_2}) \|z_2\|^2$$
$$= z_1^T \frac{\partial^2 f}{\partial w_1 \partial w_1^T} z_1 + z_2^T \frac{\partial^2 f}{\partial w_2 \partial w_2^T} z_2 + 2 z_1^T \frac{\partial^2 f}{\partial w_1 \partial w_2^T} z_2 - 4 + \frac{2\sqrt{2}}{\pi} + \frac{3\sqrt{2}}{2}$$
$$= \frac{3\sqrt{2}}{2} - \frac{2\sqrt{2}}{\pi} > 0$$

On the direction $z = (z_1, z_2) = \left(\frac{\sqrt{2}}{2}, -\frac{\sqrt{2}}{2}, -\frac{\sqrt{2}}{2}, \frac{\sqrt{2}}{2}\right)$,

$$z^T \nabla_R^2 f z = z^T \nabla^2 f z - (w_1^T \frac{\partial f}{\partial w_1}) \|z_1\|^2 - (w_2^T \frac{\partial f}{\partial w_2}) \|z_2\|^2$$
$$= z_1^T \frac{\partial^2 f}{\partial w_1 \partial w_1^T} z_1 + z_2^T \frac{\partial^2 f}{\partial w_2 \partial w_2^T} z_2 + 2 z_1^T \frac{\partial^2 f}{\partial w_1 \partial w_2^T} z_2 - 4 + \frac{2\sqrt{2}}{\pi} + \frac{3\sqrt{2}}{2}$$
$$= 1 - \frac{2\sqrt{2}}{\pi} + 1 - \frac{2\sqrt{2}}{\pi} - 2 - 4 + \frac{2\sqrt{2}}{\pi} + \frac{3\sqrt{2}}{2}$$
$$= \frac{3\sqrt{2}}{2} - \frac{2\sqrt{2}}{\pi} - 4 < 0$$

So this point is a saddle point.

## C.2 $\left(\frac{5\pi}{4}, \frac{5\pi}{4}\right)$

On the direction $z = (z_1, z_2) = \left(\frac{\sqrt{2}}{2}, -\frac{\sqrt{2}}{2}, \frac{\sqrt{2}}{2}, -\frac{\sqrt{2}}{2}\right)$,

$$w_1^T \frac{\partial f}{\partial w_1} = 2 - \frac{\sqrt{2}}{\pi} - \frac{\sqrt{2}}{4}$$

$$w_2^T \frac{\partial f}{\partial w_2} = 2 - \frac{\sqrt{2}}{\pi} - \frac{\sqrt{2}}{4}$$

$$z^T \nabla_R^2 f z = z^T \nabla^2 f z - (w_1^T \frac{\partial f}{\partial w_1}) \|z_1\|^2 - (w_2^T \frac{\partial f}{\partial w_2}) \|z_2\|^2$$

$$= z_1^T \frac{\partial^2 f}{\partial w_1 \partial w_1^T} z_1 + z_2^T \frac{\partial^2 f}{\partial w_2 \partial w_2^T} z_2 + 2z_1^T \frac{\partial^2 f}{\partial w_1 \partial w_2^T} z_2 - 4 + \frac{2\sqrt{2}}{\pi} + \frac{\sqrt{2}}{2}$$

$$= 1 + 1 + 2 - 4 + \frac{2\sqrt{2}}{\pi} + \frac{\sqrt{2}}{2}$$

$$= \frac{2\sqrt{2}}{\pi} + \frac{\sqrt{2}}{2} > 0$$

On the direction $z = (z_1, z_2) = \left(\frac{\sqrt{2}}{2}, -\frac{\sqrt{2}}{2}, -\frac{\sqrt{2}}{2}, \frac{\sqrt{2}}{2}\right)$,

$$z^T \nabla_R^2 f z = z_1^T \frac{\partial^2 f}{\partial w_1 \partial w_1^T} z_1 + z_2^T \frac{\partial^2 f}{\partial w_2 \partial w_2^T} z_2 + 2z_1^T \frac{\partial^2 f}{\partial w_1 \partial w_2^T} z_2 - 4 + \frac{2\sqrt{2}}{\pi} + \frac{\sqrt{2}}{2}$$

$$= 1 - \frac{2\sqrt{2}}{\pi} + 1 - \frac{2\sqrt{2}}{\pi} - 2 - 4 + \frac{2\sqrt{2}}{\pi} + \frac{\sqrt{2}}{2}$$

$$= \frac{\sqrt{2}}{2} - \frac{2\sqrt{2}}{\pi} - 4 < 0$$

So this point is a saddle point.

## C.3 $\left(\frac{\pi}{4}, \frac{5\pi}{4}\right)$

On the direction $z = (z_1, z_2) = \left(\frac{\sqrt{2}}{2}, -\frac{\sqrt{2}}{2}, \frac{\sqrt{2}}{2}, -\frac{\sqrt{2}}{2}\right)$,

$$w_1^T \frac{\partial f}{\partial w_1} = 1 - \frac{\sqrt{2}}{\pi} - \frac{3\sqrt{2}}{4}$$

$$w_2^T \frac{\partial f}{\partial w_2} = 1 - \frac{\sqrt{2}}{\pi} - \frac{\sqrt{2}}{4}$$

$$z^T \nabla_R^2 f z = z^T \nabla^2 f z - (w_1^T \frac{\partial f}{\partial w_1}) \|z_1\|^2 - (w_2^T \frac{\partial f}{\partial w_2}) \|z_2\|^2$$

$$= z_1^T \frac{\partial^2 f}{\partial w_1 \partial w_1^T} z_1 + z_2^T \frac{\partial^2 f}{\partial w_2 \partial w_2^T} z_2 + 2z_1^T \frac{\partial^2 f}{\partial w_1 \partial w_2^T} z_2 - 2 + \frac{2\sqrt{2}}{\pi} + \sqrt{2}$$

$$= 1 + 1 + 2 - 2 + \frac{2\sqrt{2}}{\pi} + \sqrt{2}$$

$$= 2 + \frac{2\sqrt{2}}{\pi} + \sqrt{2} > 0$$

On the direction $z = (z_1, z_2) = \left(\frac{\sqrt{2}}{2}, -\frac{\sqrt{2}}{2}, -\frac{\sqrt{2}}{2}, \frac{\sqrt{2}}{2}\right)$,

$$z^T \nabla_R^2 f z = z_1^T \frac{\partial^2 f}{\partial w_1 \partial w_1^T} z_1 + z_2^T \frac{\partial^2 f}{\partial w_2 \partial w_2^T} z_2 + 2 z_1^T \frac{\partial^2 f}{\partial w_1 \partial w_2^T} z_2 - 2 + \frac{2\sqrt{2}}{\pi} + \sqrt{2}$$

$$= 1 - \frac{2\sqrt{2}}{\pi} + 1 - \frac{2\sqrt{2}}{\pi} - 2 - 2 + \frac{2\sqrt{2}}{\pi} + \sqrt{2}$$

$$= \sqrt{2} - \frac{2\sqrt{2}}{\pi} - 2 < 0$$

So this point is a saddle point.

## C.4 CONCLUSION

In conclusion, we have four critical points: one is global maximal, the other three are saddle points.

## D  3D CASES

### D.1 WHY WE ONLY NEED 3 DIMENSION

**Lemma D.1.** *If $(w_1, w_2)$ is a critical point, then there exists a set of standard orthogonal basis $(e_1, e_2, e_3)$ such that $e_1 = w_1^*$, $e_2 = w_2^*$ and $w_1, w_2$ lies in $span\{e_1, e_2, e_3\}$.*

*Proof.* If $(w_1, w_2)$ is a critical point, then

$$(I - w_1 w_1^T) \frac{\partial f}{\partial w_1} = 0. \tag{193}$$

where matrix $(I - w_1 w_1^T)$ projects a vector onto the tangent space of $w_1$. Since

$$(I - w_1 w_1^T) w_1 = w_1 - w_1 = 0, \tag{194}$$

we get

$$(I - w_1 w_1^T) \frac{\partial f}{\partial w_1} \tag{195}$$

$$= \frac{1}{\pi} (I - w_1 w_1^T) \left( (\pi - \theta_{w_1, w_2}) w_2 - (\pi - \theta_{w_1, w_1^*}) w_1^* - (\pi - \theta_{w_1, w_2^*}) w_2^* \right), \tag{196}$$

which means that $(\pi - \theta_{w_1, w_2}) w_2 - (\pi - \theta_{w_1, w_1^*}) w_1^* - (\pi - \theta_{w_1, w_2^*}) w_2^*$ lies in the direction of $w_1$. If $\theta_{w_1, w_2} = \pi$, i.e., $w_1 = -w_2$, then of course the four vectors have rank at most 3, so we can find the proper basis. If $\theta_{w_1, w_2} < \pi$, then we know that there exists a real number $r$ such that

$$(\pi - \theta_{w_1, w_2}) w_2 - (\pi - \theta_{w_1, w_1^*}) w_1^* - (\pi - \theta_{w_1, w_2^*}) w_2^* + r \cdot w_1 = 0. \tag{197}$$

Since $\theta_{w_1, w_2} < \pi$, we know that the four vectors $w_1$, $w_2$, $w_1^*$ and $w_2^*$ are linear dependent. Thus, they have rank at most 3 and we can find the proper basis. $\square$

### D.2 SOME PROPERTIES OF CRITICAL POINTS

Next we will focus on the properties of critical points. Assume $(w_1, w_2)$ is one of the critical points, from lemma D.1 we can find a set of standard orthogonal basis $(e_1, e_2, e_3)$ such that $e_1 = w_1^*$, $e_2 = w_2^*$ and $w_1, w_2$ lies in $span\{e_1, e_2, e_3\}$. Furthermore, assume $w_1 = w_{11} e_1 + w_{12} e_2 + w_{13} e_3$ and $w_2 = w_{21} e_1 + w_{22} e_2 + w_{23} e_3$, i.e., $w_1 = (w_{11}, w_{12}, w_{13})$ and $w_2 = (w_{21}, w_{22}, w_{23})$. Since we have already found out all the critical points when $w_{13} = w_{23} = 0$, in the following we assume $w_{13}^2 + w_{23}^2 \neq 0$.

**Lemma D.2.** $\theta_{w_1, w_2} < \pi$.

*Proof.* If $\theta_{w_1, w_2} = \pi$, then $w_1 = -w_2$, so $w_2$ is in the direction of $w_1$. We have already known from (196) that $(\pi - \theta_{w_1, w_2}) w_2 - (\pi - \theta_{w_1, w_1^*}) w_1^* - (\pi - \theta_{w_1, w_2^*}) w_2^*$ lies in the direction of $w_1$, so further we know $(\pi - \theta_{w_1, w_1^*}) w_1^* + (\pi - \theta_{w_1, w_2^*}) w_2^*$ lies in the direction of $w_1$. However, $(\pi -$

$\theta_{w_1,w_1^*})w_1^* - (\pi - \theta_{w_1,w_2^*})w_2^*$ lies in $span\{e_1, e_2\}$, so $w_1 \in span\{e_1, e_2\}$ and $w_2 \in span\{e_1, e_2\}$. Thus, $w_{13} = w_{23} = 0$ and that contradicts with the assumption.

In a word, $\theta_{w_1,w_2} < \pi$. $\qquad\square$

**Lemma D.3.** $w_{13} * w_{23} \neq 0$.

*Proof.* We have already known from (196) that $(\pi - \theta_{w_1,w_2})w_2 - (\pi - \theta_{w_1,w_1^*})w_1^* - (\pi - \theta_{w_1,w_2^*})w_2^*$ lies in the direction of $w_1$. Writing it in each dimension and we know that there exists a real number $r_0$ such that

$$(\pi - \theta_{w_1,w_2})w_{21} - (\pi - \theta_{w_1,w_1^*}) = r_0 \cdot w_{11} \tag{198}$$

$$(\pi - \theta_{w_1,w_2})w_{22} - (\pi - \theta_{w_1,w_2^*}) = r_0 \cdot w_{12} \tag{199}$$

$$(\pi - \theta_{w_1,w_2})w_{23} = r_0 \cdot w_{13}. \tag{200}$$

From lemma D.2 we know that $\theta_{w_1,w_2} < \pi$, so we can define

$$k = \frac{r_0}{\pi - \theta_{w_1,w_2}}. \tag{201}$$

Then the equations become

$$w_{21} - \frac{\pi - \theta_{w_1,w_1^*}}{\pi - \theta_{w_1,w_2}} = k \cdot w_{11} \tag{202}$$

$$w_{22} - \frac{\pi - \theta_{w_1,w_2^*}}{\pi - \theta_{w_1,w_2}} = k \cdot w_{12} \tag{203}$$

$$w_{23} = k \cdot w_{13}. \tag{204}$$

Similarly, we have

$$w_{11} - \frac{\pi - \theta_{w_2,w_1^*}}{\pi - \theta_{w_1,w_2}} = k' \cdot w_{21} \tag{205}$$

$$w_{12} - \frac{\pi - \theta_{w_2,w_2^*}}{\pi - \theta_{w_1,w_2}} = k' \cdot w_{22} \tag{206}$$

$$w_{13} = k' \cdot w_{23}. \tag{207}$$

Since $w_{13}^2 + w_{23}^2 \neq 0$, at least one of those two variables cannot be 0. WLOG, we assume that $w_{13} \neq 0$. If $w_{23} = 0$, then from (207) we know that $w_{13} \neq 0$, which contradicts the assumption. Thus, $w_{23} \neq 0$, which means that $w_{13} * w_{23} \neq 0$. $\qquad\square$

**Lemma D.4.** $w_{13} * w_{23} < 0$.

*Proof.* Adapting from the proof of lemma D.3, we know that

$$w_{21} - \frac{\pi - \theta_{w_1,w_1^*}}{\pi - \theta_{w_1,w_2}} = k \cdot w_{11} \tag{208}$$

$$w_{22} - \frac{\pi - \theta_{w_1,w_2^*}}{\pi - \theta_{w_1,w_2}} = k \cdot w_{12} \tag{209}$$

$$w_{23} = k \cdot w_{13} \tag{210}$$

and

$$w_{11} - \frac{\pi - \theta_{w_2,w_1^*}}{\pi - \theta_{w_1,w_2}} = k' \cdot w_{21} \tag{211}$$

$$w_{12} - \frac{\pi - \theta_{w_2,w_2^*}}{\pi - \theta_{w_1,w_2}} = k' \cdot w_{22} \tag{212}$$

$$w_{13} = k' \cdot w_{23}. \tag{213}$$

Furthermore, $kk' = \frac{w_{23}}{w_{13}} \cdot \frac{w_{13}}{w_{23}} = 1$, so $k' = \frac{1}{k}$.

From lemma D.2 we know that $\theta_{w_1,w_2} < \pi$, and from lemma D.3 we know that both $w_1$ and $w_2$ are outside $span\{w_1^*, w_2^*\}$, so $\forall i, j \in [2], \theta_{w_i,w_j^*} < \pi$. Thus, $\forall i, j \in [2], \frac{\pi - \theta_{w_i,w_j^*}}{\pi - \theta_{w_1,w_2}} > 0$. Therefore, we have

$$w_{21} > k \cdot w_{11} \tag{214}$$

$$w_{11} > \frac{1}{k} w_{21}. \tag{215}$$

That means $k < 0$, so $\frac{w_{23}}{w_{13}} > 0$.

In a word, $w_{13} * w_{23} < 0$. $\qquad\square$

**Lemma D.5.**

$$\frac{\arccos(-w_{11})}{\arccos(-w_{21})} = \frac{\arccos(-w_{12})}{\arccos(-w_{22})} = -\frac{w_{23}}{w_{13}}. \tag{216}$$

*Proof.* Adapting from the proof of lemma D.4 and we know that

$$\frac{w_{21} - \frac{\pi - \theta_{w_1,w_1^*}}{\pi - \theta_{w_1,w_2}}}{w_{11}} = \frac{w_{22} - \frac{\pi - \theta_{w_1,w_2^*}}{\pi - \theta_{w_1,w_2}}}{w_{12}} = \frac{w_{23}}{w_{13}} = k. \tag{217}$$

Similarly, we have

$$\frac{w_{11} - \frac{\pi - \theta_{w_2,w_1^*}}{\pi - \theta_{w_1,w_2}}}{w_{21}} = \frac{w_{12} - \frac{\pi - \theta_{w_2,w_2^*}}{\pi - \theta_{w_1,w_2}}}{w_{22}} = \frac{w_{13}}{w_{23}} = \frac{1}{k}. \tag{218}$$

Taking the first component of (217) and (218) gives us

$$w_{21} = k \cdot w_{11} + \frac{\pi - \theta_{w_1,w_1^*}}{\pi - \theta_{w_1,w_2}} \tag{219}$$

$$w_{21} = k \cdot w_{11} - k \frac{\pi - \theta_{w_2,w_1^*}}{\pi - \theta_{w_1,w_2}}. \tag{220}$$

Thus,

$$\frac{\pi - \theta_{w_1,w_1^*}}{\pi - \theta_{w_2,w_1^*}} = -k. \tag{221}$$

Similarly, we get

$$\frac{\pi - \theta_{w_1,w_2^*}}{\pi - \theta_{w_2,w_2^*}} = -k. \tag{222}$$

Since $\forall i, j \in [2], \pi - \theta_{w_i,w_j^*} = \arccos(-\theta_{w_{ij}})$, we know that

$$\frac{\arccos(-w_{11})}{\arccos(-w_{21})} = \frac{\arccos(-w_{12})}{\arccos(-w_{22})} = -\frac{w_{23}}{w_{13}}. \tag{223}$$

$\qquad\square$

For simplicity, based on D.5, we define $k_0 = -k$, $\theta_1 = \pi - \theta_{w_2,w_1^*}$ and $\theta_2 = \pi - \theta_{w_2,w_2^*}$. Then

$$\pi - \theta_{w_1,w_1^*} = k_0 \theta_1 \tag{224}$$

$$\pi - \theta_{w_1,w_2^*} = k_0 \theta_2. \tag{225}$$

WLOG, assume $k_0 \geq 1$, otherwise we can switch $w_1$ and $w_2$.

Thus,

$$w_{11} = -\cos(k_0 \theta_1) \tag{226}$$

$$w_{12} = -\cos(k_0 \theta_2) \tag{227}$$

$$w_{21} = -\cos(\theta_1) \tag{228}$$

$$w_{22} = -\cos(\theta_2). \tag{229}$$

**Lemma D.6.** $\theta_1 + \theta_2 \geq \frac{\pi}{2}$.

*Proof.* Since $\theta_1 = \pi - \theta_{w_2, w_1^*}$ and $\theta_2 = \pi - \theta_{w_2, w_2^*}$, we know that $\theta_1, \theta_2 \in [0, \pi]$. Besides,

$$w_{11}^2 + w_{12}^2 = 1 - w_{13}^2 \leq 1 \tag{230}$$

$$w_{21}^2 + w_{22}^2 = 1 - w_{23}^2 \leq 1. \tag{231}$$

Thus,

$$\cos^2(k_0 \theta_1) + \cos^2(k_0 \theta_2) \leq 1 \tag{232}$$

$$\cos^2(\theta_1) + \cos^2(\theta_2) \leq 1. \tag{233}$$

If one of $\theta_1$ and $\theta_2$ is larger than $\frac{\pi}{2}$, say $\theta_1 > \frac{\pi}{2}$, then of course $\theta_1 + \theta_2 \geq \frac{\pi}{2}$. If $\theta_1, \theta_2 \in [0, \frac{\pi}{2}]$, then

$$\sin^2 \left( \frac{\pi}{2} - \theta_1 \right) = \cos^2(\theta_1) \leq 1 - \cos^2(\theta_2) = \sin^2(\theta_2), \tag{234}$$

so $\frac{\pi}{2} - \theta_1 \leq \theta_2$, which means that $\theta_1 + \theta_2 \geq \frac{\pi}{2}$.

In a word, $\theta_1 + \theta_2 \geq \frac{\pi}{2}$. $\qquad\square$

**Lemma D.7.** $1 \leq k_0 \leq 3$.

*Proof.* First we prove that $k_0 \leq 4$: From lemma D.6, we know that $\theta_1 + \theta_2 \geq \frac{\pi}{2}$, so at least one of $\theta_1$ and $\theta_2$ is no less than $\frac{\pi}{4}$, say $\theta_1 \geq \frac{\pi}{4}$. If $k_0 > 4$, then $\pi - \theta_{w_1, w_1^*} = k_0 \theta_1 > \pi$, which makes a contradiction. Thus, $k_0 \leq 4$.

Furthermore, if $3 < k_0 \leq 4$, then $\theta_1, \theta_2 \in [0, \frac{\pi}{3}]$ because $k_0 \theta_1, k_0 \theta_2 \in [0, \pi]$.

If $\theta_1, \theta_2 \in [0, \frac{\pi}{4})$, then $\theta_1 + \theta_2 < \frac{\pi}{2}$ which contradicts lemma D.6.

If $\theta_1, \theta_2 \in [\frac{\pi}{4}, \frac{\pi}{3}]$, then $k_0 \theta_1, k_0 \theta_2 \in (\frac{3\pi}{4}, \pi]$, which means that $\cos^2(k_0 \theta_1) + \cos^2(k_0 \theta_2) > \frac{1}{2} + \frac{1}{2} = 1$ and contradicts (232).

If $\theta_1 \leq \frac{\pi}{4} \leq \theta_2$ and $k_0 \theta_1 < \frac{\pi}{2}$, then $\theta_1 < \frac{\pi}{2k_0} < \frac{\pi}{6}$, so from lemma D.6, $\theta_2 \geq \frac{\pi}{2} - \theta_1 > \frac{\pi}{3}$, which contradicts $k_0 \theta_2 \leq \pi$.

If $\theta_1 \leq \frac{\pi}{4} \leq \theta_2$ and $k_0 \theta_1 \geq \frac{\pi}{2}$, then $k_0 \theta_1, k_0 \theta_2 \in [\frac{\pi}{2}, \pi]$. Since $\cos^2(k_0 \theta_1) + \cos^2(k_0 \theta_2) \leq 1$, we know that

$$\sin^2 \left( k_0 \theta_1 - \frac{\pi}{2} \right) = \cos^2(k_0 \theta_1) \leq 1 - \cos^2(k_0 \theta_2) = \sin^2(\pi - k_0 \theta_2), \tag{235}$$

so $k_0 \theta_1 - \frac{\pi}{2} \leq \pi - k_0 \theta_2$, which means that $k_0 \theta_1 + k_0 \theta_2 \leq \frac{3\pi}{2}$. Thus, $\theta_1 + \theta_2 < \frac{\pi}{2}$, which contradicts lemma D.6.

In a word, $1 \leq k_0 \leq 3$. $\qquad\square$

**Lemma D.8.** *Define*

$$F(\theta) = \frac{-k_0 \theta}{k_0 \cos(k_0 \theta) + \cos(\theta)}, \tag{236}$$

*then* $F(\theta_1) = F(\theta_2) (\theta_1, \theta_2 \in [0, \frac{\pi}{k_0}])$.

*Proof.* Since $k_0 \theta_1, k_0 \theta_2 \in [0, \pi]$, we know that $\theta_1, \theta_2 \in [0, \frac{\pi}{k_0}]$.

From (217), applying the change of variables on the first component and we get

$$\frac{-\cos \theta_1 - \frac{k_0 \theta_1}{\pi - \theta_{w_1, w_2}}}{-\cos(k_0 \theta_1)} = -k_0. \tag{237}$$

Thus,

$$\pi - \theta_{w_1, w_2} = \frac{-k_0 \theta_1}{k_0 \cos(k_0 \theta_1) + \cos(\theta_1)} = F(\theta_1). \tag{238}$$

Similarly, if we apply the change of variables onto the second component of (217), we will get

$$\pi - \theta_{w_1,w_2} = \frac{-k_0\theta_2}{k_0\cos(k_0\theta_2) + \cos(\theta_2)} = F(\theta_2). \tag{239}$$

Thus,

$$F(\theta_1) = F(\theta_2)(\theta_1, \theta_2 \in [0, \frac{\pi}{k_0}]). \tag{240}$$

$\square$

**Lemma D.9.** $\exists \theta_0 \in [\frac{\pi}{2k_0}, \frac{3\pi}{4k_0})$, s.t.,

$$F(\theta) = \begin{cases} < 0 & 0 \le \theta < \theta_0 \\ = \infty & \theta = \theta_0 \\ > 0 & \theta_0 < \theta \le \frac{\pi}{k_0} \end{cases} . \tag{241}$$

*Proof.* Note that when $\theta \in [0, \frac{\pi}{k_0}]$, $-k_0\theta$ is always non-positive. Define $G(\theta) = k_0\cos(k_0\theta) + \cos(\theta)$, then $G(\theta)$ is a strict decreasing function w.r.t. $\theta$. Note that $G(0) = k_0 + 1 > 0$ and $G\left(\frac{\pi}{k_0}\right) = \cos\left(\frac{\pi}{k_0}\right) - k_0 < 0$, so there must be an $\theta_0 \in (0, \frac{\pi}{k_0})$ such that $G(\theta_0) = 0$. Thus, when $0 \le \theta < \theta_0$, $G(\theta) > 0$, and when $\theta_0 \le \frac{\pi}{k_0}$, $G(\theta) < 0$.

Thus,

$$F(\theta) = \begin{cases} < 0 & 0 \le \theta < \theta_0 \\ = \infty & \theta = \theta_0 \\ > 0 & \theta_0 < \theta \le \frac{\pi}{k_0} \end{cases} . \tag{242}$$

Then the only thing we need to prove is $\frac{\pi}{2k_0} \le \theta_0 < \frac{3\pi}{4k_0}$. Note that

$$G(\frac{\pi}{2k_0}) = \cos\left(\frac{\pi}{2k_0}\right) \ge 0 \tag{243}$$

$$G(\frac{3\pi}{4k_0}) = \cos\left(\frac{3\pi}{4k_0}\right) - \frac{k_0}{\sqrt{2}} \le \frac{\sqrt{2}}{2} - \frac{\sqrt{2}}{2} = 0. \tag{244}$$

Since the inequality (244) holds only when $\cos\left(\frac{3\pi}{4k_0}\right) = \frac{\sqrt{2}}{2}$ and $\frac{k_0}{\sqrt{2}} = \frac{\sqrt{2}}{2}$, which means $k_0 = 3$ and $k_0 = 1$, which makes a contradiction. Thus,

$$G(\frac{3\pi}{4k_0}) < 0. \tag{245}$$

Therefore, $\frac{\pi}{2k_0} \le \theta_0 < \frac{3\pi}{4k_0}$, which completes the proof. $\square$

**Lemma D.10.** $F(\theta)$ is either strictly decreasing or first decrease and then increase when $\theta \in (\theta_0, \frac{\pi}{k_0}]$.

*Proof.*

$$F'(\theta) = -\frac{k_0\left(k_0\cos(k_0\theta) + \cos(\theta)\right) - k_0\theta\left(-k_0^2\sin(k_0\theta) - \sin\theta\right)}{\left(k_0\cos(k_0\theta) + \cos(\theta)\right)^2} \tag{246}$$

$$= -k_0\frac{k_0\cos(k_0\theta) + \cos\theta + k_0^2\theta\sin(k_0\theta) + \theta\sin\theta}{\left(k_0\cos(k_0\theta) + \cos(\theta)\right)^2}. \tag{247}$$

Define $H(\theta) = k_0\cos(k_0\theta) + \cos\theta + k_0^2\theta\sin(k_0\theta) + \theta\sin\theta(\theta \in (\theta_0, \frac{\pi}{k_0}])$, then $H(\theta) \cdot F'(\theta) < 0$(i.e., when $H(\theta)$ is positive, $F(\theta)$ is decreasing, otherwise $F(\theta)$ is increasing), and we know that

$$H'(\theta) = -k_0^2\sin(k_0\theta) - \sin\theta + k_0^3\theta\cos(k_0\theta) + k_0^2\sin(k_0\theta) + \theta\cos\theta + \sin\theta \tag{248}$$

$$= k_0^3\theta\cos(k_0\theta) + \theta\cos\theta \tag{249}$$

$$= \theta(k_0^3\cos(k_0\theta) + \cos\theta) \tag{250}$$

$$\le \theta(k_0\cos(k_0\theta) + \cos\theta) \tag{251}$$

$$= \theta \cdot G(\theta) \tag{252}$$

$$< 0. \tag{253}$$

Note that (251) holds because $\theta > \theta_0 \geq \frac{\pi}{2k_0}$.

Thus, $H(\theta)$ is a strictly decreasing function when $\theta \in (\theta_0, \frac{\pi}{k_0}]$.

We can see that

$$H(\theta_0) = G(\theta_0) + k_0^2 \theta_0 \sin(k_0 \theta_0) + \theta_0 \sin \theta_0 \tag{254}$$

$$= k_0^2 \theta_0 \sin(k_0 \theta_0) + \theta_0 \sin \theta_0 > 0. \tag{255}$$

Thus, if $H(\frac{\pi}{k_0}) \geq 0$, then $F(\theta)$ is monotonically decreasing when $\theta \in (\theta_0, \frac{\pi}{k_0}]$. Otherwise, $F(\theta)$ first decrease and then increase when $\theta \in (\theta_0, \frac{\pi}{k_0}]$. □

**Lemma D.11.** $\forall \theta \in (\frac{3\pi}{4k_0}, \frac{\pi}{k_0}], F(\theta) < F\left(\frac{3\pi}{4k_0}\right)$.

*Proof.* From lemma D.10 we have already known that $F(\theta)$ is either strictly decreasing or first decrease and then increase when $\theta \in (\theta_0, \frac{\pi}{k_0}]$, so the maximum of the function value on an interval can only be at the endpoints of that interval, which means that we only need to prove $F(\frac{3\pi}{4k_0}) > F(\frac{\pi}{k_0})$.

Note that

$$F(\frac{3\pi}{4k_0}) > F(\frac{\pi}{k_0}) \tag{256}$$

$$\Leftrightarrow \frac{\frac{3\pi}{4}}{\frac{\sqrt{2}}{2}k_0 - \cos\left(\frac{3\pi}{4k_0}\right)} > \frac{\pi}{k_0 - \cos\frac{\pi}{k_0}} \tag{257}$$

$$\Leftrightarrow \frac{\frac{3}{4}}{\frac{\sqrt{2}}{2}k_0 - \cos\left(\frac{3\pi}{4k_0}\right)} > \frac{1}{k_0 - \cos\frac{\pi}{k_0}} \tag{258}$$

$$\Leftrightarrow \frac{3}{4}\left(k_0 - \cos\frac{\pi}{k_0}\right) > \frac{\sqrt{2}}{2}\left(k_0 - \cos\left(\frac{3\pi}{4k_0}\right)\right) \tag{259}$$

$$\Leftrightarrow \left(\frac{3}{4} - \frac{\sqrt{2}}{2}\right)k_0 > \frac{3}{4}\cos\frac{\pi}{k_0} - \cos\left(\frac{3\pi}{4k_0}\right). \tag{260}$$

Let $h(x) = \frac{3}{4}\cos x - \cos\left(\frac{3x}{4}\right) (x \in [\frac{\pi}{3}, \pi])$, then

$$h'(x) = \frac{3}{4}\left(\sin\left(\frac{3x}{4}\right) - \sin x\right). \tag{261}$$

Thus, $h(x)$ is decreasing in $[\frac{\pi}{3}, \frac{4\pi}{7}]$ and increasing in $[\frac{4\pi}{7}, \pi]$. However, we know that $h(\frac{\pi}{3}) = \frac{3}{8} - \frac{\sqrt{2}}{2} < 0$ and $h(\pi) = -\frac{3}{4} + \frac{\sqrt{2}}{2} < 0$, so $h(x)$ is negative when $x \in [\frac{\pi}{3}, \pi]$.

Therefore,

$$\left(\frac{3}{4} - \frac{\sqrt{2}}{2}\right)k_0 > 0 > \frac{3}{4}\cos\frac{\pi}{k_0} - \cos\left(\frac{3\pi}{4k_0}\right), \tag{262}$$

which means that $F(\frac{3\pi}{4k_0}) > F(\frac{\pi}{k_0})$.

Thus, $\forall \theta \in (\frac{3\pi}{4k_0}, \frac{\pi}{k_0}], F(\theta) < F\left(\frac{3\pi}{4k_0}\right)$. □

**Lemma D.12.** $\theta_1 = \theta_2$.

*Proof.* From the proof of lemma D.8 we get

$$F(\theta_1) = \pi - \theta_{w_1, w_2} = F(\theta_2). \tag{263}$$

Thus, $F(\theta_1), F(\theta_2) \in [0, \pi]$.

Using lemma D.9, $\theta_1, \theta_2 > \theta_0 \geq \frac{\pi}{2k_0}$, so that $k_0 \theta_1, k_0 \theta_2 \in (\frac{\pi}{2}, \pi]$.

From (232), we know that $k_0(\theta_1 + \theta_2) \leq \frac{3\pi}{2}$, which means that at least one of $\theta_1$ and $\theta_2$ are less than or equal to $\frac{3\pi}{4k_0}$, w.l.o.g. we assume $\theta_1 \leq \frac{3\pi}{4k_0}$.

Note that lemma D.11 tells us that $F(\frac{3\pi}{4k_0}) > F(\frac{\pi}{k_0})$, so at the point $\theta = \frac{3\pi}{4k_0}$, the function cannot be increasing, which combining with lemma D.10 shows that $F(\theta)$ is strictly decreasing when $\theta \in (\theta_0, \frac{3\pi}{4k_0}]$.

If $\theta_2 > \frac{3\pi}{4k_0}$, then we know that $F(\theta_1) \geq F\left(\frac{3\pi}{4k_0}\right) > F(\theta_2)$, which contradicts $F(\theta_1) = F(\theta_2)$.

Thus, $\theta_1, \theta_2(\theta_0, \frac{3\pi}{4k_0}]$. Since $F(\theta)$ is monotonically decreasing when $\theta \in (\theta_0, \frac{3\pi}{4k_0}]$, we can conclude that $\theta_1 = \theta_2$. $\qquad \square$

## D.3 NEGATIVE CURVATURE

First we compute the Hessian matrix:

If $z = (tz_1, z_2)$, $||z_1|| = ||z_2|| = 1$ and $w_1^T z_1 = w_2^T z_2 = 0$, then

$$z^T \nabla_R^2 f z = \tag{264}$$

$$t^2 \left( \frac{1}{\pi\sqrt{1 - (w_1^T w_2)^2}} \left(z_1^T w_2\right)^2 - \frac{1}{\pi\sqrt{1 - (w_1^T w_1^*)^2}} \left(z_1^T w_1^*\right)^2 - \frac{1}{\pi\sqrt{1 - (w_1^T w_2^*)^2}} \left(z_1^T w_2^*\right)^2 \right) \tag{265}$$

$$+ \frac{1}{\pi\sqrt{1 - (w_1^T w_2)^2}} \left(z_2^T w_1\right)^2 - \frac{1}{\pi\sqrt{1 - (w_2^T w_1^*)^2}} \left(z_2^T w_1^*\right)^2 - \frac{1}{\pi\sqrt{1 - (w_2^T w_2^*)^2}} \left(z_2^T w_2^*\right)^2 \tag{266}$$

$$+ t \left( \frac{2}{\pi\sqrt{1 - (w_1^T w_2)^2}} z_1^T w_2 w_1^T z_2 + \frac{2}{\pi}(\pi - \theta_{w_1,w_2}) z_1^T z_2 \right) \tag{267}$$

$$- t^2 \left( \frac{1}{\pi}(\pi - \theta_{w_1,w_2}) w_1^T w_2 - \frac{1}{\pi}(\pi - \theta_{w_1,w_1^*}) w_1^T w_1^* - \frac{1}{\pi}(\pi - \theta_{w_1,w_2^*}) w_1^T w_2^* \right) \tag{268}$$

$$- \left( \frac{1}{\pi}(\pi - \theta_{w_1,w_2}) w_2^T w_1 - \frac{1}{\pi}(\pi - \theta_{w_2,w_1^*}) w_2^T w_1^* - \frac{1}{\pi}(\pi - \theta_{w_2,w_2^*}) w_2^T w_2^* \right). \tag{269}$$

**Lemma D.13.** *For every critical point $(w_1, w_2)$ outside $span\{w_1^*, w_2^*\}$,*

$$\frac{1}{\pi}(\pi - \theta_{w_1,w_2}) w_1^T w_2 - \frac{1}{\pi}(\pi - \theta_{w_1,w_1^*}) w_1^T w_1^* - \frac{1}{\pi}(\pi - \theta_{w_1,w_2^*}) w_1^T w_2^* \tag{270}$$

$$= -k_0(\pi - \theta_{w_1,w_2}) \tag{271}$$

$$\frac{1}{\pi}(\pi - \theta_{w_1,w_2}) w_2^T w_1 - \frac{1}{\pi}(\pi - \theta_{w_2,w_1^*}) w_2^T w_1^* - \frac{1}{\pi}(\pi - \theta_{w_2,w_2^*}) w_2^T w_2^* \tag{272}$$

$$= -\frac{1}{k_0}(\pi - \theta_{w_1,w_2}). \tag{273}$$

*Proof.* In lemma D.3, we have three equations, and we write them again for convenience:

$$(\pi - \theta_{w_1,w_2}) w_{21} - (\pi - \theta_{w_1,w_1^*}) = r_0 \cdot w_{11} \tag{274}$$

$$(\pi - \theta_{w_1,w_2}) w_{22} - (\pi - \theta_{w_1,w_2^*}) = r_0 \cdot w_{12} \tag{275}$$

$$(\pi - \theta_{w_1,w_2}) w_{23} = r_0 \cdot w_{13}. \tag{276}$$

Multiply 274 by $w_{11}$, 275 by $w_{12}$, 276 by $w_{13}$, we get

$$(\pi - \theta_{w_1,w_2}) w_{21} w_{11} - (\pi - \theta_{w_1,w_1^*}) w_{11} = r_0 \cdot w_{11}^2 \tag{277}$$

$$(\pi - \theta_{w_1,w_2}) w_{22} w_{12} - (\pi - \theta_{w_1,w_2^*}) w_{12} = r_0 \cdot w_{12}^2 \tag{278}$$

$$(\pi - \theta_{w_1,w_2}) w_{23} w_{13} = r_0 \cdot w_{13}^2. \tag{279}$$

Combine these three equations, we know that

$$\frac{1}{\pi}(\pi - \theta_{w_1,w_2})w_1^T w_2 - \frac{1}{\pi}(\pi - \theta_{w_1,w_1^*})w_1^T w_1^* - \frac{1}{\pi}(\pi - \theta_{w_1,w_2^*})w_1^T w_2^* \tag{280}$$

$$= r_0 \tag{281}$$

$$= (\pi - \theta_{w_1,w_2})\frac{w_{23}}{w_{13}} \tag{282}$$

$$= -k_0(\pi - \theta_{w_1,w_2}). \tag{283}$$

Similarly,

$$\frac{1}{\pi}(\pi - \theta_{w_1,w_2})w_2^T w_1 - \frac{1}{\pi}(\pi - \theta_{w_2,w_1^*})w_2^T w_1^* - \frac{1}{\pi}(\pi - \theta_{w_2,w_2^*})w_2^T w_2^* \tag{284}$$

$$= (\pi - \theta_{w_1,w_2})\frac{w_{13}}{w_{23}} \tag{285}$$

$$= -\frac{1}{k_0}(\pi - \theta_{w_1,w_2}). \tag{286}$$

$\square$

**Lemma D.14.** *For every critical point $(w_1, w_2)$ outside $span\{w_1^*, w_2^*\}$, there is negative curvature.*

*Proof.* We select $z_1 = (-\frac{\sqrt{2}}{2}, \frac{\sqrt{2}}{2}, 0)$ and $z_2 = (\frac{\sqrt{2}}{2}, -\frac{\sqrt{2}}{2}, 0)$, then

$$z^T \nabla_R^2 f z = -\frac{1}{\sqrt{1-w_{11}^2}} - \frac{1}{\sqrt{1-w_{21}^2}} + \left(k_0 + \frac{1}{k_0} - 2\right)(\pi - \theta_{w_1,w_2}). \tag{287}$$

From lemma D.7 we know that $1 \le k_0 \le 3$.

If $1 \le k_0 \le 2$, then

$$z^T \nabla_R^2 f z \le -2 + \frac{1}{2} \cdot \pi < 0. \tag{288}$$

If $2 < k_0 \le 3$, from (232) and lemma D.12 we get $2\cos^2(k_0\theta_1) \le 1$, so $k_0\theta_1 \le \frac{3\pi}{4}$, which means that

$$\theta_1 \le \frac{3\pi}{4k_0} < \frac{3\pi}{8}. \tag{289}$$

Thus,

$$|w_{11}| = |\cos\theta_1| > \cos\frac{3\pi}{8}. \tag{290}$$

Besides, from (233) and lemma D.12 we know that $2\cos^2\theta_1 \le 1$, so $\theta_1 \ge \frac{\pi}{4}$, which means that

$$k_0\theta_1 > 2 \cdot \frac{\pi}{4} = \frac{\pi}{2}. \tag{291}$$

Using (289) and (291),

$$w_{11} = w_{12} = -\cos(k_0\theta_1) > 0 \tag{292}$$

$$w_{21} = w_{22} = -\cos\theta_1 < 0. \tag{293}$$

From lemma D.4, we conclude that

$$\langle w_1, w_2 \rangle = w_{11} \cdot w_{21} + w_{12} \cdot w_{22} + w_{13} \cdot w_{23} < 0, \tag{294}$$

which means that

$$\theta_{w_1,w_2} > \frac{\pi}{2}. \tag{295}$$

Thus,

$$z^T \nabla_R^2 f z \le -\frac{1}{\sqrt{1-\cos^2\frac{3\pi}{8}}} - 1 + \left(3 + \frac{1}{3} - 2\right)\left(\pi - \frac{\pi}{2}\right) \tag{296}$$

$$= -\sqrt[4]{2} - 1 + \frac{2\pi}{3} \tag{297}$$

$$< 0. \tag{298}$$

In a word, for every critical point $(w_1, w_2)$ outside $span\{w_1^*, w_2^*\}$, there is negative curvature. $\square$

# E   2D CASES WITH ASSUMPTION RELAXATION

Since this section is pretty similar to B, I will try my best to make it brief and point out the most important things in the proof.

## E.1   PRELIMINARIES

After the changing of variables(i.e., polar coordinates), we know that $w_1 = (\cos\theta_1, \sin\theta_1)$ and $w_2 = (\cos\theta_2, \sin\theta_2)$. And the manifold gradient(expressed by $m$) are $m(w_1) = \sin\theta_1 \frac{\partial f}{\partial w_{11}} - \cos\theta_1 \frac{\partial f}{\partial w_{12}}$ and $m(w_2) = \sin\theta_2 \frac{\partial f}{\partial w_{21}} - \cos\theta_2 \frac{\partial f}{\partial w_{22}}$.

Applying the changing of variables and multiply it by $\pi$, we get

$$m(w_1) = (\pi - \theta_{w_1,w_2})\sin(\theta_1 - \theta_2) + (\pi - \theta_{w_1,w_2^*})\sin(\alpha - \theta_1) - (\pi - \theta_{w_1,w_1^*})\sin\theta_1. \quad (299)$$

And

$$m(w_2) = (\pi - \theta_{w_1,w_2})\sin(\theta_2 - \theta_1) + (\pi - \theta_{w_2,w_2^*})\sin(\alpha - \theta_2) - (\pi - \theta_{w_2,w_1^*})\sin\theta_2. \quad (300)$$

Define(where $w = (\cos\theta, \sin\theta)$)

$$h(\theta) = (\pi - \theta_{w,w_2^*})\sin(\alpha - \theta) - (\pi - \theta_{w,w_1^*})\sin\theta. \quad (301)$$

Then when $\theta$ is in the first part to the fourth part, the function $h$ will change to four different functions:

$$h_1(\theta) = (\pi - \alpha + \theta)\sin(\alpha - \theta) - (\pi - \theta)\sin\theta \quad (302)$$

$$h_2(\theta) = (\pi - \theta + \alpha)\sin(\alpha - \theta) - (\pi - \theta)\sin\theta \quad (303)$$

$$h_3(\theta) = (\pi - \theta + \alpha)\sin(\alpha - \theta) - (\theta - \pi)\sin\theta \quad (304)$$

$$h_4(\theta) = (\theta - \alpha - \pi)\sin(\alpha - \theta) - (\pi - \theta)\sin\theta. \quad (305)$$

WLOG, we assume $\theta_1 \leq \theta_2$.

## E.2   $0 \leq \theta_1 \leq \theta_2 \leq \alpha$

First, it's easy to verify that $\forall \theta \in [0, \theta], h_1(\theta) + h_1(\alpha - \theta) = 0$.

Besides,

$$h_1'(\theta) = \sin\theta + \sin(\alpha - \theta) - (\pi - \theta)\cos\theta - (\pi - \alpha + \theta)\cos(\alpha - \theta) \quad (306)$$

$$= 2\sin\frac{\alpha}{2}\cos(\theta - \frac{\alpha}{2}) - (\pi - \theta)\cos\theta - (\pi - \alpha + \theta)\cos(\alpha - \theta) \quad (307)$$

$$\leq 2\sin\frac{\alpha}{2} - \frac{\pi}{2}(\cos\theta + \cos(\alpha - \theta)) \quad (308)$$

$$= 2\sin\frac{\alpha}{2} - \pi\cos\frac{\alpha}{2}\cos(\theta - \frac{\alpha}{2}) \quad (309)$$

$$\leq 2\sin\frac{\alpha}{2} - \pi\cos\frac{\alpha}{2} \qquad < 0. \quad (310)$$

When $m(w_1) = m(w_2) = 0$, we know that $h_1(\theta_1) + h_1(\theta_2) = 0$, and because of those two properties above, we know that $\theta_1 + \theta_2 = \alpha$. Thus, $\theta_1 \in [0, \frac{\alpha}{2}]$. And we have the following lemma

**Lemma E.1.** $m(w_1) \leq 0$.

*Proof.*

$$m(w_1) = \sin(\alpha - 2\theta_1)(\pi - \alpha + 2\theta_1) - (\pi - \alpha + \theta_1)\sin(\alpha - \theta_1) + (\pi - \theta_1)\sin\theta_1 \quad (311)$$

$$\geq \sin(\alpha - 2\theta_1)(\pi - \alpha + \theta_1) - (\pi - \alpha + \theta_1)\sin(\alpha - \theta_1) + (\pi - \theta_1)\sin\theta_1 \quad (312)$$

$$\geq \sin(\alpha - 2\theta_1)(\pi - \alpha + \theta_1) - (\pi - \alpha + \theta_1)\sin(\alpha - \theta_1) + (\pi - \frac{\alpha}{2})\sin\theta_1 \quad (313)$$

$$= (\pi - \alpha + \theta_1)(\sin(\alpha - 2\theta_1) - \sin(\alpha - \theta_1)) + (\pi - \frac{\alpha}{2})\sin\theta_1 \quad (314)$$

$$\geq (\pi - \frac{\alpha}{2})(\sin(\alpha - 2\theta_1) - \sin(\alpha - \theta_1) + \sin\theta_1) \quad (315)$$

$$= (\pi - \frac{\alpha}{2})(\sin(\alpha - 2\theta_1) - \sin\theta_1 - \sin\theta_1\cos(\alpha - 2\theta_1) - \cos\theta_1\sin(\alpha - 2\theta_1)) \quad (316)$$

$$\geq 0. \quad (317)$$

Thus, the only possible critical points are $m(w_1) = 0$, which are $0$ and $\frac{\alpha}{2}$. After verification, we conclude that there are only two critical points in this case: $(\theta_1, \theta_2) = (0, \alpha)$ or $(\theta_1, \theta_2) = (\frac{\alpha}{2}, \frac{\alpha}{2})$. $\qquad\square$

## E.3 $\quad \alpha \leq \theta_1 \leq \theta_2 \leq \pi$

When $m(w_1) = m(w_2) = 0$, we know that $h_1(\theta_1) + h_1(\theta_2) = 0$. However, when $\theta \in [\alpha, \pi]$, we know that

$$h_2(\theta) = (\pi - \theta + \alpha)\sin(\alpha - \theta) - (\pi - \theta)\sin\theta \qquad\qquad \leq 0. \qquad (318)$$

The inequality cannot become equal because the possible values of $\theta$s such that each term equals zero has no intersection. Thus, $h_2(\theta)$ is always negative, which means that in this case there are no critical points.

## E.4 $\quad \pi \leq \theta_1 \leq \theta_2 \leq \pi + \alpha$

It's easy to verify that $\forall \theta \in [\pi, \pi + \alpha], h_3(\theta) + h_3(2\pi + \alpha - \theta) = 0$. Furthermore,

$$h_3'(\theta) = -\sin(\alpha - \theta) - \cos(\alpha - \theta)(\pi + \alpha - \theta) - \sin\theta - (\theta - \pi)\cos\theta \qquad (319)$$

$$= -2\sin\frac{\alpha}{2}\cos(\theta - \frac{\alpha}{2}) - (\theta - \pi)\cos\theta - (\pi + \alpha - \theta)\cos(\alpha - \theta) \qquad (320)$$

$$> 0. \qquad (321)$$

Thus, from $m(w_1) = m(w_2) = 0$, we know that $h_1(\theta_1) + h_1(\theta_2) = 0$ we get $\theta_1 + \theta_2 = 2\pi + \alpha$, which means that $\theta_1 \in [\pi, \pi + \frac{\alpha}{2}]$, so we can prove the following lemma:

**Lemma E.2.** $m(w_1) \leq 0$.

*Proof.* Let $\theta' = \theta_1 - \pi$, then

$$m(w_1) = (\pi - \theta_2 + \theta_1)\sin(\theta_1 - \theta_2) + h_3(\theta_1) \qquad (322)$$

$$= (\pi + \theta' - \alpha + \theta')\sin(2\theta' - \alpha) + h_1(\theta') + \pi\sin\theta' - \pi\sin(\alpha - \theta') \qquad (323)$$

$$\leq (\pi + 2\theta' - \alpha)\sin(2\theta' - \alpha) + \sin(\alpha - 2\theta')(\pi + 2\theta' - \alpha) + \pi(\sin\theta' - \sin(\alpha - \theta')) \qquad (324)$$

$$\leq \pi(\sin\theta' - \cos\theta') \qquad (325)$$

$$\leq 0. \qquad (326)$$

The first inequality is from lemma E.1. $\qquad\square$

Thus, the only possible critical points are $m(w_1) = 0$, which are $\pi$ and $\pi + \frac{\alpha}{2}$. After verification, we conclude that there are only two critical points in this case: $(\theta_1, \theta_2) = (\pi, \pi + \alpha)$ or $(\theta_1, \theta_2) = (\pi + \frac{\alpha}{2}, \pi + \frac{\alpha}{2})$.

