# OpenReview forum: "No Spurious Local Minima in a Two Hidden Unit ReLU Network"
_ICLR.cc/2018/Conference — Invite to Workshop Track_

### Official Review · AnonReviewer1 · 2017-11-18
**Due to poor presentation and clarity of ideas, as well as description of how this work differs from previous ones, I believe that the paper needs major revision before publication. So for the moment, rejection.**

**Rating:** 4
**Confidence:** 4

**Review:**

Summary:
The paper considers the problem of a single hidden layer neural network, with 2 RELU units (this is what I got from the paper - as I describe below, it was not clear at all the setting of the problem - if I'm mistaken, I will also wait for the rest of the reviews to have a more complete picture of the problem).
Given this architecture, the authors focus on characterizing the objective landscape of such a problem.
The techniques used depend on previous work. According to the authors, this paper extends(?) previous results on a NN with a single layer with a single unit.

Originality:
The paper heavily depends on the approach followed by Brutzkus and Globerson, 2017. To this end, slighly novel.

Importance:
Understanding the landscape (local vs global minima vs saddle points) is an important direction in order to further understand when and why deep neural networks work. I would say that the topic is an important one.

Presentation/Clarity:
To the best of my understanding, the paper has some misconceptions. The title is not clear whether the paper considers a two layer RELU network or a single layer with with two RELU units. In the abstract the authors state that it has to do with a two-layer RELU network with two hidden units (per layer? in total?). Later on, in Section 3, the expression at the bottom of page 2 seems to consider a single-layer RELU network, with two units.
These are crucial for understanding the contribution of the paper; while reading the paper, I assumed that the authors consider the case of a single hidden unit with K = 2 RELU activations (however, that complicated my understanding on how it compares with state of the art).

Another issue is the fact that, on my humble opinion, the main text looks like a long proof. It would be great to have more intuitions.

Comments:
1. The paper mainly focuses on a specific problem instance, where the weight vectors are unit-normed and orthogonal to each other. While the authors already identify that this might be a restriction, it still does not lessen the fact that the configuration considered is a really specific one.

2. The paper reads like a collection of lemmas, with no verbose connection. It was hard to read and understand their value, just because mostly the text was structured as one lemma after the other.

3. It is not clear from the text whether the setting is already considered in Brutzkus and Globerson, 2017. Please clarify how your work is different/new from previous works.

---

> ### Author Response · Authors · 2017-12-23
> **Response**
>
> We thank the reviewers for their thoughtful comments. We believe that our paper makes a meaningful contribution to understanding the global properties of neural networks, and in particular to understanding gradient-based optimization in deep learning. Although our setting is simple, it is already significantly more complicated than other works that analyze global geometry of neural networks, which have mostly focused on the case of linear networks or networks with a single filter.
>
> 1.	We apologize for the confusion of the setting and model from the abstract. The abstract has been updated. As stated clearly in the first displayed equation of the paper (beginning of Section 3), the architecture in this paper is y= \sigma(<w_1,x>) +\sigma(<w_2,x>).
>
> 2.	The only thing we use from Brutzkus and Globerson, 2017 is is the formula: “E[relu(w^T x) relu (v^Tx)]=  formula” and the rest is novel. In Brutzkus and Globerson, 2017, they consider one-hidden-layer network with No Overlap. Due to this assumption, they can identify every critical point in their network. However, in our problem, it’s impossible to identify every critical point and we use a different method to analyze the landscape of the network. Our analysis is substantially different from Brutzkus and Globerson 2017 and this setting is NOT considered in Brutzkus and Globerson.
> 3.	We have submitted a revision that clarifies our contributions and adds more intuition.

---

### Official Review · AnonReviewer2 · 2017-11-26
**No Spurious Local Minima in a Two Hidden Unit ReLU Network**

**Rating:** 6
**Confidence:** 3

**Review:**

In this paper the authors studied the theoretical properties of manifold descent approaches in a standard regression problem, whose regressor is a simple neural network. Leveraged by two recent results in global optimization, they showed that with a simple two-layer ReLU network with two hidden units, the problem with a standard MSE population loss function does not have spurious local minimum points. Based on the results by Lee et al, which shows that first order methods converge to local minimum solution (instead of saddle points), it can be concluded that the global minima of this problem can be found by any manifold descent techniques, including standard gradient descent methods. In general I found this paper clearly written and technically sound. I also appreciate the effort of developing theoretical results for deep learning, even though the current results are restrictive to very simple NN architectures.

Contribution:
As discussed in the literature review section, apart from previous results that studied the theoretical convergence properties for problems that involves a single hidden unit NN, this paper extends the convergence results to problems that involves NN with two hidden units. The analysis becomes considerably more complicated, and the contribution seems to be novel and significant. I am not sure why did the authors mentioned the work on over-parameterization though. It doesn't seem to be relevant to the results of this paper (because the NN architecture proposed in this paper is rather small).

Comments on the Assumptions:
- Please explain the motivation behind the standard Gaussian assumption of the input vector x.
- Please also provide more motivations regarding the assumption of the orthogonality of weights: w_1^\top w_2=0 (or the acute angle assumption in Section 6).
Without extra justifications, it seems that the theoretical result only holds for an artificial problem setting. While the ReLU activation is very common in NN architecture, without more motivations I am not sure what are the impacts of these results.

General Comment:
The technical section is quite lengthy, and unfortunately I am not available to go over every single detail of the proofs. From the analysis in the main paper, I believe the theoretical contribution is correct and sound. While I appreciate the technical contributions, in order to improve the readability of this paper, it would be great to see more motivations of the problem studied in this paper (even with simple examples). Furthermore, it is important to discuss the technical assumptions on the 1) standard Gaussianity of the input vector, and 2) the orthogonality of the weights (and the acute angle assumption in Section 6) on top of the discussions in Section 8.1, as they are critical to the derivations of the main theorems.

---

> ### Author Response · Authors · 2017-12-23
> **Response**
>
> We thank the reviewers for their thoughtful comments. We believe that our paper makes a meaningful contribution to understanding the global properties of neural networks, and in particular to understanding gradient-based optimization in deep learning. Although our setting is simple, it is already significantly more complicated than other works that analyze global geometry of neural networks, which have mostly focused on the case of linear networks or networks with a single filter.
>
> 1.	Brutzkus and Globerson, 2017 showed learning No-Overlap Networks without some distributional assumption is NP-hard. However, the No-Overlap Networks can be also be learned for Gaussian Inputs. Following Brutzkus and Globerson 2017 and Tian 2017, we also make the Gaussian assumption on the input vector in our model.
> 2.	We agree that we have a lot of conditions on the architecture of the network. Most of these conditions are used to simplify the proof, which are very involved even after these simplifications. However, we do believe that the conclusions hold more broadly or approximately hold (meaning gradient descent finds local optima that are nearly globally optimal), but we are unable to prove this now and leave it as a future work.

---

### Official Review · AnonReviewer3 · 2017-11-26
**A proof of a conjecture of the landscape of the objective value in deep learning**

**Rating:** 6
**Confidence:** 2

**Review:**

This paper considers a special deep learning model and shows that in expectation, there is only one unique local minimizer. As a result, a gradient descent algorithm converges to the unique solution. This works address a conjecture proposed by Tian (2017).

While it is clearly written, my main concern is whether this model is significant enough. The assumptions K=2 and v1=v2=1 reduces the difficulty of the analysis, but it makes the model considerably simpler than any practical setting.

---

> ### Author Response · Authors · 2017-12-23
> **Response**
>
> We thank the reviewers for their thoughtful comments. We believe that our paper makes a meaningful contribution to understanding the global properties of neural networks, and in particular to understanding gradient-based optimization in deep learning. Although our setting is simple, it is already significantly more complicated than other works that analyze global geometry of neural networks, which have mostly focused on the case of linear networks or networks with a single filter.
>
> 1.	We agree that the architecture is simpler than any practical setting. However, this simple architecture of two hidden units is already significantly more complicated than many previous works that analyze global geometry of neural networks, which focus on the case of linear networks or networks with a single filter.

---

### Decision · Program_Chairs · 2018-01-29
**ICLR 2018 Conference Acceptance Decision**

**Decision:**

Invite to Workshop Track

**Comment:**

This submission is a continuation of a line of theoretical work that seeks to characterize optimization landscapes of neural networks by the presence or absence of spurious local minima.  As the number of critical points grows combinatorially for larger networks, it is very challenging to show such results.  The present submission extends slightly previous work by considering two hidden units and their proof technique goes beyond that of Brutzkus and Globerson, 2017, potentially leading to more interesting results if they can be extended to more complex networks.

The setting of two hidden units is quite limited - far from any practical setting.  If this were the stepping stone to proving optimality of certain optimization strategies for more complex networks, this may be of some interest, but it seems doubtful.  One indication is given in Sec. 7 / Fig. 1 in which it is shown that for even quite small numbers of hidden units, spurious local optima do occur and are reached 40% of the time for random initializations even with only 11 nodes.